# DIVERGENCE-REGULARIZED MULTI-AGENT ACTOR-CRITIC

## ABSTRACT

Entropy regularization is a popular method in reinforcement learning (RL). Although it has many advantages, it alters the RL objective and makes the converged policy deviate from the optimal policy of the original Markov Decision Process. Though divergence regularization has been proposed to settle this problem, it cannot be trivially applied to cooperative multi-agent reinforcement learning (MARL). In this paper, we investigate divergence regularization in cooperative MARL and propose a *novel* off-policy cooperative MARL framework, divergence-regularized multi-agent actor-critic (DMAC). Mathematically, we derive the update rule of DMAC which is naturally off-policy, guarantees a monotonic policy improvement and is *not* biased by the regularization. DMAC is a flexible framework and can be combined with many existing MARL algorithms. We evaluate DMAC in a didactic stochastic game and StarCraft Multi-Agent Challenge and empirically show that DMAC substantially improves the performance of existing MARL algorithms.

## 1 INTRODUCTION

Regularization is a common method for single-agent reinforcement learning (RL). The optimal policy learned by traditional RL algorithm is always deterministic (Sutton and Barto, 2018). This property may result in the inflexibility of the policy facing with unknown environments (Yang et al., 2019). Entropy regularization is proposed to settle this problem by learning a policy according to the maximum-entropy principle (Haarnoja et al., 2017). Moreover, entropy regularization is beneficial to exploration and robustness for RL algorithms (Haarnoja et al., 2018). However, entropy regularization is imperfect. Eysenbach and Levine (2019) pointed out maximum-entropy RL is a modification of the original RL objective because of the entropy regularizer. Maximum-entropy RL is actually learning an optimal policy for the entropy-regularized Markov Decision Process (MDP) rather than the original MDP, *i.e.*, the converged policy may be biased. Nachum et al. (2017) analysed a more general case for regularization in RL and proposed what we call *divergence regularization*. Divergence regularization can avoid the bias of the converged policy as well as be beneficial to exploration. Wang et al. (2019) employed divergence regularizer and proposed a single-agent RL algorithm, DAPO, which prevents the altering-objective drawback of entropy regularization.

Regularization can also be applied to cooperative multi-agent reinforcement learning (MARL) (Agarwal et al., 2020; Zhang et al., 2021). However, most of cooperative MARL algorithms do not use regularizer (Lowe et al., 2017; Foerster et al., 2018; Rashid et al., 2018; Son et al., 2019; Jiang et al., 2020; Wang et al., 2021a). Only few cooperative MARL algorithms such as FOP (Zhang et al., 2021) use entropy regularization, which may suffer from the drawback aforementioned. Divergence regularization, on the other hand, could potentially benefit cooperative MARL. In addition to its advantages mentioned above, divergence regularization can also help to control the step size of policy update which is similar to conservative policy iteration (Kakade and Langford, 2002) in single-agent RL. Conservative policy iteration and its successive methods such as TRPO (Schulman et al., 2015) and PPO (Schulman et al., 2017) can stabilize policy improvement (Touati et al., 2020). These methods use a surrogate objective for policy update, but decentralized policies in centralized training with decentralized execution (CTDE) paradigm may not preserve the properties of the surrogate objective. Moreover, DAPO (Wang et al., 2019) cannot be trivially extended to cooperative MARL settings. Even with some tricks like V-trace (Espeholt et al., 2018) for off-policy correction, DAPO is essentially an on-policy algorithm and thus may not be sample-efficient in cooperative MARL settings.

In the paper, we propose and analyze *divergence policy iteration* in general cooperative MARL settings and a special case combined with value decomposition. Based on divergence policy iteration, we derive the off-policy update rule for the critic, policy, and target policy and propose divergence-regularized multi-agent actor-critic (DMAC), a *novel* off-policy cooperative MARL framework. We theoretically show that DMAC guarantees a monotonic policy improvement and is *not* biased by the regularization. Besides, DMAC is beneficial to exploration and stable policy improvement by applying our update rule of target policy. Moreover, DMAC is a flexible framework and can be combined with many existing cooperative MARL algorithms to substantially improve their performance.

We empirically investigate DMAC in a didactic stochastic game and StarCraft Multi-Agent Challenge (Samvelyan et al., 2019). We combine DMAC with five representative MARL methods, *i.e.*, COMA (Foerster et al., 2018) for on-policy multi-agent policy gradient, MAAC (Iqbal and Sha, 2019) for off-policy multi-agent actor-critic, QMIX (Rashid et al., 2018) for value decomposition, DOP (Wang et al., 2021b) for the combination of value decomposition and policy gradient, and FOP (Zhang et al., 2021) for the combination of value decomposition and entropy regularization. Experimental results show that DMAC indeed induces better performance, faster convergence, and better stability in most tasks, which verifies the benefits of DMAC and demonstrates the advantages of divergence regularization over entropy regularization in cooperative MARL.

## 2 RELATED WORK

**MARL.** MARL has been a hot topic in the field of RL. In this paper, we focus on cooperative MARL. Cooperative MARL is usually modeled as Dec-POMDP (Oliehoek et al., 2016), where all agents share a reward and aim to maximize the long-term return. Centralized training with decentralized execution (CTDE) (Lowe et al., 2017) paradigm is widely used in cooperative MARL. CTDE usually utilizes a centralized value function to address the non-stationarity for multi-agent settings and decentralized policies for scalability. Many MARL algorithms adopt CTDE paradigm such as COMA, MAAC, QMIX, DOP and FOP. COMA (Foerster et al., 2018) employs the counterfactual baseline which can reduce the variance as well as settle the credit assignment problem. MAAC (Iqbal and Sha, 2019) uses self-attention mechanism to integrate local observation and action of each agent and provides the structured information for the centralized critic. Value decomposition (Sunehag et al., 2018; Rashid et al., 2018; Son et al., 2019; Yang et al., 2020; Wang et al., 2021a;b; Zhang et al., 2021) is a popular class of cooperative MARL algorithms. These methods express the global Q-function as a function of individual Q-functions to satisfy Individual-Global-Max (IGM), which means the optimal actions of individual Q-functions are corresponding to the optimal joint action of global Q-function. QMIX (Rashid et al., 2018) is a representative of value decomposition methods. It uses a hypernet to ensure the monotonicity of the global Q-function in terms of individual Q-functions, which is a sufficient condition of IGM. DOP (Wang et al., 2021b) is a method that combines value decomposition with policy gradient. DOP uses a linear value decomposition which is another sufficient condition of IGM and the linear value decomposition helps the compute of policy gradient. FOP (Zhang et al., 2021) is a method that combines value decomposition with entropy regularization and uses a more general condition, Individual-Global-Optimal, to replace IGM. In this paper, we will combine DMAC with these algorithms and show its improvement.

**Regularization.** Entropy regularization was first proposed in single-agent RL. Nachum et al. (2017) analyzed the entropy-regularized MDP and revealed the properties about the optimal policy and the corresponding Q-function and V-function. They also showed the equivalence of value-based methods and policy-based methods in entropy-regularized MDP. Haarnoja et al. (2018) pointed out maximum-entropy RL can achieve better exploration and stability facing with the model and estimation error. Although entropy regularization has many advantages, Eysenbach and Levine (2019) showed entropy regularization modifies the MDP and results in the bias of the convergent policy. Yang et al. (2019) revealed the drawbacks of the convergent policy of general RL and maximum-entropy RL. The former is usually a deterministic policy (Sutton and Barto, 2018) which is not flexible enough for unknown situations, while the latter is a policy with non-zero probability for all actions which may be dangerous in some scenarios. Neu et al. (2017) analyzed the entropy regularization method from several views. They revealed a more general form of regularization which is actually divergence regularization and showed entropy regularization is just a special case of divergence regularization. Wang et al. (2019) absorbed previous result and proposed an on-policy algorithm, *i.e.*, DAPO. However, DAPO cannot be trivially applied to MARL. Moreover, its on-policy learning is

not sample-efficient for MARL settings and its off-policy correction trick V-trace (Espeholt et al., 2018) is also intractable in MARL. There are some previous studies in single-agent RL which use similar target policy to ours, but their purposes are quite different. Trust-PCL (Nachum et al., 2018) introduces a target policy as a trust region constraint for maximum-entropy RL, but the policy is still biased by entropy regularizer. MIRL (Grau-Moya et al., 2019) uses a distribution which is only related to actions as the target policy to compute a mutual-information regularizer, but it still changes the objective of the original RL.

## 3 PRELIMINARIES

**Dec-POMDP** is a general model for cooperative MARL. A Dec-POMDP is a tuple $M = \{S, A, P, Y, O, I, n, r, \gamma\}$. $S$ is the state space, $n$ is the number of agents, $\gamma$ is the discount factor, and $I = \{1, 2 \cdots n\}$ is the set of all agents. $A = A_1 \times A_2 \times \cdots \times A_n$ represents the joint action space where $A_i$ is the individual action space for agent $i$. $P(s'|s, \boldsymbol{a}) : S \times A \times S \to [0, 1]$ is the transition function, and $r(s, \boldsymbol{a}) : S \times A \to \mathbb{R}$ is the reward function of state $s$ and joint action $\boldsymbol{a}$. $Y$ is the observation space, and $O(s, i) : S \times I \to Y$ is a mapping from state to observation for each agent. The objective of Dec-POMDP is to maximize $J(\boldsymbol{\pi}) = \mathbb{E}_{\boldsymbol{\pi}} \left[ \sum_{t=0} \gamma^t r(s_t, \boldsymbol{a}_t) \right]$, and thus we need to find the optimal joint policy $\boldsymbol{\pi}^* = \arg\max_{\boldsymbol{\pi}} J(\boldsymbol{\pi})$. To settle the partial observable problem, history $\tau_i \in \mathcal{T}_i = (Y \times A_i)^*$ is often used to replace observation $o_i \in Y$. As for policies in CTDE, each agent $i$ has an individual policy $\pi_i(a_i|\tau_i)$ and the joint policy $\boldsymbol{\pi}$ is the product of each $\pi_i$. Though we calculate individual policy as $\pi_i(a_i|\tau_i)$ in practice, we will use $\pi_i(a_i|s)$ in analysis and proofs for simplicity.

**Entropy regularization** adds the logarithm of current policy to the reward function. It modifies the optimization objective as $J_{\text{ent}}(\boldsymbol{\pi}) = \mathbb{E}_{\boldsymbol{\pi}} \left[ \sum_{t=0} \gamma^t \left( r(s_t, \boldsymbol{a}_t) - \lambda \log \boldsymbol{\pi}(\boldsymbol{a}_t|s_t) \right) \right]$. We also have the corresponding Q-function $Q_{\text{ent}}^{\boldsymbol{\pi}}(s, \boldsymbol{a}) = r(s, \boldsymbol{a}) + \gamma \mathbb{E} \left[ V_{\text{ent}}^{\boldsymbol{\pi}}(s') \right]$ and V-function $V_{\text{ent}}^{\boldsymbol{\pi}}(s) = \mathbb{E} \left[ Q_{\text{ent}}^{\boldsymbol{\pi}}(s, \boldsymbol{a}) - \lambda \log \boldsymbol{\pi}(\boldsymbol{a}|s) \right]$. Given these definitions, we can deduce an interesting property $V_{\text{ent}}^{\boldsymbol{\pi}}(s) = \mathbb{E} \left[ Q_{\text{ent}}^{\boldsymbol{\pi}}(s, \boldsymbol{a}) \right] + \lambda \mathcal{H} \left( \boldsymbol{\pi}(\cdot|s) \right)$, where $\mathcal{H} \left( \boldsymbol{\pi}(\cdot|s) \right)$ represents the entropy of policy $\boldsymbol{\pi}(\cdot|s)$. $V_{\text{ent}}^{\boldsymbol{\pi}}(s)$ includes an entropy term which is the reason it is called *entropy regularization*.

## 4 METHOD

In this section, we first give the definition of divergence regularization. Then we propose and analyze *divergence policy iteration*. Finally, based on divergence policy iteration, we derive the update rules of the critic, policy, and target policy for divergence-regularized MARL.

### 4.1 DIVERGENCE REGULARIZATION

We maintain a target policy $\rho_i$ for each agent $i$, which is different from the policy $\pi_i$. Then we have a joint target policy $\boldsymbol{\rho} = \prod_{i=1}^n \rho_i$. This joint target policy $\boldsymbol{\rho}$ modifies the objective function as $J_{\boldsymbol{\rho}}(\boldsymbol{\pi}) = \mathbb{E}_{\boldsymbol{\pi}} \left[ \sum_{t=0} \gamma^t \left( r(s_t, \boldsymbol{a}_t) - \lambda \log \frac{\boldsymbol{\pi}(\boldsymbol{a}_t|s_t)}{\boldsymbol{\rho}(\boldsymbol{a}_t|s_t)} \right) \right]$. That is, a regularizer $\log \frac{\boldsymbol{\pi}(\boldsymbol{a}_t|s_t)}{\boldsymbol{\rho}(\boldsymbol{a}_t|s_t)}$, which describes the *discrepancy* between policy $\boldsymbol{\pi}$ and target policy $\boldsymbol{\rho}$, is added to the reward function just like entropy regularization.

Given $\boldsymbol{\rho}$, we can define corresponding V-function and Q-function for divergence regularization as follows,

$$V_{\boldsymbol{\rho}}^{\boldsymbol{\pi}}(s) = \mathbb{E}_{\boldsymbol{\pi}} \left[ \sum_{t=0} \gamma^t (r(s_t, \boldsymbol{a}_t) - \lambda \log \frac{\boldsymbol{\pi}(\boldsymbol{a}_t|s_t)}{\boldsymbol{\rho}(\boldsymbol{a}_t|s_t)})|s_0 = s \right] \tag{1}$$

$$Q_{\boldsymbol{\rho}}^{\boldsymbol{\pi}}(s, \boldsymbol{a}) = r(s, \boldsymbol{a}) + \gamma \mathbb{E}_{s' \sim P(\cdot|s,a)} \left[ V_{\boldsymbol{\rho}}^{\boldsymbol{\pi}}(s') \right]. \tag{2}$$

Further, by simple deduction, we have

$$V_{\boldsymbol{\rho}}^{\boldsymbol{\pi}}(s) = \mathbb{E}_{\boldsymbol{a} \sim \boldsymbol{\pi}(\cdot|s)} \left[ Q_{\boldsymbol{\rho}}^{\boldsymbol{\pi}}(s, \boldsymbol{a}) - \lambda \log \frac{\boldsymbol{\pi}(\boldsymbol{a}|s)}{\boldsymbol{\rho}(\boldsymbol{a}|s)} \right] = \mathbb{E}_{\boldsymbol{a} \sim \boldsymbol{\pi}(\cdot|s)} \left[ Q_{\boldsymbol{\rho}}^{\boldsymbol{\pi}}(s, \boldsymbol{a}) \right] - \lambda D_{\text{KL}} \left( \boldsymbol{\pi}(\cdot|s) \| \boldsymbol{\rho}(\cdot|s) \right).$$

$V_{\boldsymbol{\rho}}^{\boldsymbol{\pi}}(s)$ includes an extra term which is the KL divergence between $\boldsymbol{\pi}$ and $\boldsymbol{\rho}$, and thus this regularizer is referred to as *divergence regularization*.

## 4.2  DIVERGENCE POLICY ITERATION

From the perspective of policy evaluation, we can define an operator $\Gamma_{\boldsymbol{\rho}}^{\boldsymbol{\pi}}$ as

$$\Gamma_{\boldsymbol{\rho}}^{\boldsymbol{\pi}} Q(s, \boldsymbol{a}) = r(s, \boldsymbol{a}) + \gamma \mathbb{E}_{s' \sim P(\cdot|s,\boldsymbol{a}), \boldsymbol{a}' \sim \boldsymbol{\pi}(\cdot|s')} \left[ Q(s', \boldsymbol{a}') - \lambda \log \frac{\boldsymbol{\pi}(\boldsymbol{a}'|s')}{\boldsymbol{\rho}(\boldsymbol{a}'|s')} \right] \tag{3}$$

and have the following lemma. Note that *all the proofs* are given in Appendix A.

**Lemma 1 (Divergence Policy Evaluation)** *For any initial Q-function $Q^0(s, \boldsymbol{a}) : S \times A \to \mathbb{R}$, we define a sequence $\{Q^k\}$ given operator $\Gamma_{\boldsymbol{\rho}}^{\boldsymbol{\pi}}$ as $Q^{k+1} = \Gamma_{\boldsymbol{\rho}}^{\boldsymbol{\pi}} Q^k$. Then, the sequence will converge to $Q_{\boldsymbol{\rho}}^{\boldsymbol{\pi}}$ as $k \to \infty$.*

After the evaluation of the policy, we need a method to improve the policy. We have the following lemma about policy improvement.

**Lemma 2 (Divergence Policy Improvement)** *If we define $\boldsymbol{\pi}_{\text{new}}$ satisfying*

$$\boldsymbol{\pi}_{\text{new}}(\cdot|s) = \arg \min_{\boldsymbol{\pi}} D_{\text{KL}} \left( \boldsymbol{\pi}(\cdot|s) \| \boldsymbol{u}(\cdot|s) \right), \tag{4}$$

*where $\boldsymbol{u}(\cdot|s) = \boldsymbol{\rho}(\cdot|s) \frac{\exp(Q_{\boldsymbol{\rho}}^{\boldsymbol{\pi}_{\text{old}}}(s,\cdot)/\lambda)}{Z^{\boldsymbol{\pi}_{\text{old}}}(s)}$ and $Z^{\boldsymbol{\pi}_{\text{old}}}(s)$ is a normalization term, then for all actions $\boldsymbol{a}$ and all states $s$ we have $Q_{\boldsymbol{\rho}}^{\boldsymbol{\pi}_{\text{new}}}(s, \boldsymbol{a}) \geq Q_{\boldsymbol{\rho}}^{\boldsymbol{\pi}_{\text{old}}}(s, \boldsymbol{a})$.*

Lemma 1 and 2 could be seen as corollaries of the conclusion of Haarnoja et al. (2018). Lemma 2 indicates that given a policy $\boldsymbol{\pi}_{\text{old}}$, if we find a policy $\boldsymbol{\pi}_{\text{new}}$ according to (4), then the policy $\boldsymbol{\pi}_{\text{new}}$ is better than $\boldsymbol{\pi}_{\text{old}}$.

Lemma 2 does not make any assumption and is for general settings. Further, the policy improvement can be established and simplified based on value decomposition. In the following, we give an example for linear value decomposition like DOP (*i.e.*, $Q(s, \boldsymbol{a}) = \sum_i k_i(s) Q_i(s, a_i) + b(s)$) (Wang et al., 2021b).

**Lemma 3 (Divergence Policy Improvement with Linear Value Decomposition)** *If Q-functions satisfy $Q_{\boldsymbol{\rho}}^{\boldsymbol{\pi}}(s, \boldsymbol{a}) = \sum_i k_i(s) Q_{\boldsymbol{\rho}}^{\pi_i}(s, a_i) + b(s)$ and we define $\pi_{\text{new}}^i$ satisfying*

$$\pi_{\text{new}}^i(\cdot|s) = \arg \min_{\pi_i} D_{\text{KL}} \left( \pi_i(\cdot|s) \| u_i(\cdot|s) \right) \quad \forall i \in I,$$

*where $u_i(\cdot|s) = \rho_i(\cdot|s) \frac{\exp\left( k_i(s) Q_{\boldsymbol{\rho}}^{\pi_{\text{old}}^i}(s,\cdot)/\lambda \right)}{Z^{\pi_{\text{old}}^i}(s)}$ and $Z^{\pi_{\text{old}}^i}(s)$ is a normalization term, then for all actions $\boldsymbol{a}$ and all states $s$ we have $Q_{\boldsymbol{\rho}}^{\boldsymbol{\pi}_{\text{new}}}(s, \boldsymbol{a}) \geq Q_{\boldsymbol{\rho}}^{\boldsymbol{\pi}_{\text{old}}}(s, \boldsymbol{a})$.*

Lemma 3 further tells us that if the MARL setting satisfies the linear value decomposition condition, then each agent can optimize its individual policy with an objective of its own individual Q-function, which immediately improves the joint policy. By combining divergence policy evaluation and divergence policy improvement, we have the following theorem of *divergence policy iteration*.

**Theorem 1 (Divergence Policy Iteration)** *By iteratively using Divergence Policy Evaluation and Divergence Policy Improvement, we will get a sequence $\{Q^k\}$ and this sequence will converge to the optimal Q-function $Q_{\boldsymbol{\rho}}^*$ and the corresponding policy sequence will converge to the optimal policy $\boldsymbol{\pi}_{\boldsymbol{\rho}}^*$.*

Theorem 1 shows that with repeated application of divergence policy improvement and divergence policy evaluation, the policy can be *monotonically* improved and converge to the optimal policy. Let $\boldsymbol{\pi}_{\boldsymbol{\rho}}^*$, $V_{\boldsymbol{\rho}}^*(s)$, and $Q_{\boldsymbol{\rho}}^*(s, \boldsymbol{a})$ denote the optimal policy, Q-function, and V-function respectively, given a target policy $\boldsymbol{\rho}$. We have the following proposition.

**Proposition 1** *If $\boldsymbol{\pi}_{\boldsymbol{\rho}}^* = \arg \max_{\boldsymbol{\pi}} J_{\boldsymbol{\rho}}(\boldsymbol{\pi})$, and $V_{\boldsymbol{\rho}}^*(s) = V_{\boldsymbol{\rho}}^{\boldsymbol{\pi}_{\boldsymbol{\rho}}^*}(s)$ and $Q_{\boldsymbol{\rho}}^*(s, \boldsymbol{a}) = Q_{\boldsymbol{\rho}}^{\boldsymbol{\pi}_{\boldsymbol{\rho}}^*}(s, \boldsymbol{a})$ are respectively the corresponding Q-function and V-function of $\boldsymbol{\pi}_{\boldsymbol{\rho}}^*$, then they satisfy the following*

*properties:*

$$\boldsymbol{\pi}_{\boldsymbol{\rho}}^*(\boldsymbol{a}|s) \propto \boldsymbol{\rho}(\boldsymbol{a}|s) \exp\left(\left(r(s,\boldsymbol{a}) + \gamma\mathbb{E}_{s'\sim P(\cdot|s,\boldsymbol{a})}\left[V_{\boldsymbol{\rho}}^*(s')\right]\right)/\lambda\right) \tag{5}$$

$$V_{\boldsymbol{\rho}}^*(s) = \lambda\log\sum_{\boldsymbol{a}}\boldsymbol{\rho}(\boldsymbol{a}|s)\exp\left(\left(r(s,\boldsymbol{a}) + \gamma\mathbb{E}_{s'\sim P(\cdot|s,\boldsymbol{a})}\left[V_{\boldsymbol{\rho}}^*(s')\right]\right)/\lambda\right) \tag{6}$$

$$Q_{\boldsymbol{\rho}}^*(s,\boldsymbol{a}) = r(s,\boldsymbol{a}) + \gamma\lambda\mathbb{E}_{s'\sim P(\cdot|s,\boldsymbol{a})}\left[\log\sum_{\boldsymbol{a}'}\boldsymbol{\rho}(\boldsymbol{a}|s)\exp\left(Q_{\boldsymbol{\rho}}^*(s',\boldsymbol{a}')/\lambda\right)\right]. \tag{7}$$

With all these results above, we have enough tools to obtain the practical update rule of the critic, policy and target policy of DMAC.

### 4.3 DIVERGENCE-REGULARIZED CRITIC

From Proposition 1, we can obtain

$$\boldsymbol{\pi}_{\boldsymbol{\rho}}^*(\boldsymbol{a}|s) = \frac{\boldsymbol{\rho}(\boldsymbol{a}|s)\exp\left(\left(r(s,\boldsymbol{a}) + \gamma\mathbb{E}_{s'\sim P(\cdot|s,\boldsymbol{a})}\left[V_{\boldsymbol{\rho}}^*(s')\right]\right)/\lambda\right)}{\sum_{\boldsymbol{b}}\boldsymbol{\rho}(\boldsymbol{b}|s)\exp\left(\left(r(s,\boldsymbol{b}) + \gamma\mathbb{E}_{s'\sim P(\cdot|s,\boldsymbol{b})}\left[V_{\boldsymbol{\rho}}^*(s')\right]\right)/\lambda\right)} = \frac{\boldsymbol{\rho}(\boldsymbol{a}|s)\exp\left(Q_{\boldsymbol{\rho}}^*(s,\boldsymbol{a})/\lambda\right)}{\exp\left(V_{\boldsymbol{\rho}}^*(s)/\lambda\right)}$$
$$= \boldsymbol{\rho}(\boldsymbol{a}|s)\exp\left(\left(Q_{\boldsymbol{\rho}}^*(s,\boldsymbol{a}) - V_{\boldsymbol{\rho}}^*(s)\right)/\lambda\right). \tag{8}$$

By rearranging the equation, we have

$$V_{\boldsymbol{\rho}}^*(s) = Q_{\boldsymbol{\rho}}^*(s,\boldsymbol{a}) - \lambda\log\frac{\boldsymbol{\pi}_{\boldsymbol{\rho}}^*(\boldsymbol{a}|s)}{\boldsymbol{\rho}(\boldsymbol{a}|s)}, \tag{9}$$

which is tenable for all actions $\boldsymbol{a} \in A$. Therefore, we have the following corollary.

**Corollary 1**

$$Q_{\boldsymbol{\rho}}^*(s,\boldsymbol{a}) = r(s,\boldsymbol{a}) + \gamma\mathbb{E}_{s'\sim P(\cdot|s,\boldsymbol{a}),\boldsymbol{a}'\sim\boldsymbol{\pi}_{\boldsymbol{\rho}}^*(\cdot|s')}\left[Q_{\boldsymbol{\rho}}^*(s',\boldsymbol{a}') - \lambda\log\frac{\boldsymbol{\pi}_{\boldsymbol{\rho}}^*(\boldsymbol{a}'|s')}{\boldsymbol{\rho}(\boldsymbol{a}'|s')}\right]$$

*is tenable for all actions $\boldsymbol{a}' \in A$.*

Corollary 1 gives an iterative formula for $Q_{\boldsymbol{\rho}}^*(s,\boldsymbol{a})$, with which we can design a loss function and update rule for learning the critic,

$$\mathcal{L}_Q = \mathbb{E}\left[\left(Q_\phi(s,\boldsymbol{a}) - y\right)^2\right], \text{ where } y = r(s,\boldsymbol{a}) + \gamma\left(Q_{\tilde{\phi}}(s',\boldsymbol{a}') - \lambda\log\frac{\boldsymbol{\pi}(\boldsymbol{a}'|s')}{\boldsymbol{\rho}(\boldsymbol{a}'|s')}\right), \tag{10}$$

where $\phi$ and $\tilde{\phi}$ are respectively the weights of Q-function and target Q-function. The update of Q-function is similar to that in general MDP, except that the action for next state could be chosen *arbitrarily* while it must be the action that maximizes Q-function for next state in general MDP. This property greatly enhances the flexibility of learning Q-function, *e.g.*, we can easily extend it to TD($\lambda$).

### 4.4 DIVERGENCE-REGULARIZED ACTORS

DAPO (Wang et al., 2019) analyzes the divergence-regularized MDP from the perspective of policy gradient theorem (Sutton et al., 2000) and gives an on-policy update rule for single-agent RL. Unlike existing work, we focus on a different perspective and derive an off-policy update rule by taking into consideration the characteristics of MARL.

From Lemma 2, we can obtain an optimization target for policy improvement,

$$\arg\min_{\boldsymbol{\pi}} D_{\mathrm{KL}}\left(\boldsymbol{\pi}(\cdot|s)\|\boldsymbol{\rho}(\cdot|s)\frac{\exp\left(Q_{\boldsymbol{\rho}}^{\boldsymbol{\pi}}(s,\cdot)/\lambda\right)}{Z^{\boldsymbol{\pi}}(s)}\right) = \arg\max_{\boldsymbol{\pi}}\sum_{\boldsymbol{a}}\boldsymbol{\pi}(\boldsymbol{a}|s)\left(Q_{\boldsymbol{\rho}}^{\boldsymbol{\pi}}(s,\boldsymbol{a}) - \lambda\log\frac{\boldsymbol{\pi}(\boldsymbol{a}|s)}{\boldsymbol{\rho}(\boldsymbol{a}|s)}\right).$$

Then, we can define the objective of the actors,

$$\mathcal{L}_{\boldsymbol{\pi}} = \mathbb{E}_{s\sim\mathcal{D}}\left[\sum_{\boldsymbol{a}}\boldsymbol{\pi}(\boldsymbol{a}|s)\left(Q_{\boldsymbol{\rho}}^{\boldsymbol{\pi}}(s,\boldsymbol{a}) - \lambda\log\frac{\boldsymbol{\pi}(\boldsymbol{a}|s)}{\boldsymbol{\rho}(\boldsymbol{a}|s)}\right)\right], \tag{11}$$

where $\mathcal{D}$ is the replay buffer. Suppose each individual policy $\pi_i$ has a corresponding parameterization $\theta_i$. We can obtain the following policy gradient for each agent with some derivation and the detail is given in Appendix A.6,

$$\nabla_{\theta_i}\mathcal{L}_{\boldsymbol{\pi}} = \mathbb{E}_{s\sim\mathcal{D},\boldsymbol{a}\sim\boldsymbol{\pi}}\left[\nabla_{\theta_i}\log\pi_i(a_i|s)\left(Q_{\boldsymbol{\rho}}^{\boldsymbol{\pi}}(s,\boldsymbol{a}) - \lambda\log\frac{\boldsymbol{\pi}(\boldsymbol{a}|s)}{\boldsymbol{\rho}(\boldsymbol{a}|s)} - \lambda\right)\right]. \tag{12}$$

We need to point out that the key to *off-policy* update is that Lemma 2 does not limit the state distribution. It only requires the condition is satisfied for each state. Therefore, we can maintain a replay buffer to cover different states as much as possible, which is a common practice in off-policy learning. DAPO uses a similar formula to ours, but it obtains the formula from policy gradient theorem, which requires the state distribution of the current policy.

Further, we can add a counterfactual baseline to the gradient. First, we have the following equation about the counterfactual baseline (Foerster et al., 2018),

$$\mathbb{E}_{s\sim\mathcal{D},\boldsymbol{a}\sim\boldsymbol{\pi}}\left[\nabla_{\theta_i}\log\pi_i(a_i|s)b(s,a_{-i})\right]=0, \tag{13}$$

where $a_{-i}$ denotes the joint action of all agents except agent $i$. Next, we take the baseline as

$$b(s,a_{-i})=\mathbb{E}_{a_i\sim\pi_i}[(Q_{\boldsymbol{\rho}}^{\boldsymbol{\pi}}(s,\boldsymbol{a})-\lambda\log\frac{\boldsymbol{\pi}(\boldsymbol{a}|s)}{\boldsymbol{\rho}(\boldsymbol{a}|s)}-\lambda)]. \tag{14}$$

Then, the gradient for each agent $i$ can be modified as follows,

$$\nabla_{\theta_i}\mathcal{L}_{\boldsymbol{\pi}}=\mathbb{E}[\nabla_{\theta_i}\log\pi_i(a_i|s)(Q_{\boldsymbol{\rho}}^{\boldsymbol{\pi}}(s,\boldsymbol{a})-\lambda\log\frac{\boldsymbol{\pi}(\boldsymbol{a}|s)}{\boldsymbol{\rho}(\boldsymbol{a}|s)}-\lambda-b(s,a_{-i}))]$$

$$=\mathbb{E}[\nabla_{\theta_i}\log\pi_i(a_i|s)(Q_{\boldsymbol{\rho}}^{\boldsymbol{\pi}}(s,\boldsymbol{a})-\lambda\log\frac{\pi_i(a_i|s)}{\rho_i(a_i|s)}-\mathbb{E}_{a_i\sim\pi_i}[Q_{\boldsymbol{\rho}}^{\boldsymbol{\pi}}(s,\boldsymbol{a})]+\lambda D_{\mathrm{KL}}(\pi_i(\cdot|s)\|\rho_i(\cdot|s)))].$$

In addition to variance reduction and credit assignment, this counterfactual baseline eliminates the policies of other agents from the gradient. This property makes it convenient to calculate the gradient and easy to select the target policy for each agent. Moreover, if the linear value decomposition condition is satisfied, we have the following gradient formula,

$$\nabla_{\theta_i}\mathcal{L}_{\boldsymbol{\pi}}=\mathbb{E}[\nabla_{\theta_i}\log\pi_i(a_i|s)(k_i(s)A_{\boldsymbol{\rho}}^{\pi_i}(s,a_i)-\lambda\log\frac{\pi_i(a_i|s)}{\rho_i(a_i|s)}+\lambda D_{\mathrm{KL}}(\pi_i(\cdot|s)\|\rho_i(\cdot|s)))], \tag{15}$$

where $A_{\boldsymbol{\rho}}^{\pi_i}(s,a_i)=Q_{\boldsymbol{\rho}}^{\pi_i}(s,a_i)-\mathbb{E}_{\tilde{a}_i\sim\pi_i}\left[Q_{\boldsymbol{\rho}}^{\pi_i}(s,\tilde{a}_i)\right].$

### 4.5 TARGET POLICY

We have discussed the update rule of the critic and actors, and now we focus on the selection and update rule of the target policy. With the update rules above, we can obtain divergence policy iteration given a fixed target policy $\boldsymbol{\rho}$. Then we need to devise the update rule of $\boldsymbol{\rho}$ to prevent the bias of regularization and benefit the learning procedure.

Intuitively, this regularizer $\log\frac{\boldsymbol{\pi}(\boldsymbol{a}_t|s_t)}{\boldsymbol{\rho}(\boldsymbol{a}_t|s_t)}$ could help to balance exploration and exploitation. For example, for some action $\boldsymbol{a}$, if $\boldsymbol{\rho}(\boldsymbol{a}|s)>\boldsymbol{\pi}(\boldsymbol{a}|s)$, then the regularizer is equivalent to adding a positive value to the reward and *vice versa*. Therefore, if we choose previous policy as target policy, the regularizer will encourage agents to take actions whose probability has decreased and discourage agents to take actions whose probability has increased. Additionally, the regularizer actually controls the discrepancy between current policy and previous policy, which could stabilize the policy improvement (Kakade and Langford, 2002; Schulman et al., 2015; 2017).

To derive the update rule of target policy, we need to further analyze the regularizer theoretically. Let $\mu_{\boldsymbol{\pi}}$ denote the state-action distribution given a policy $\boldsymbol{\pi}$. That is, $\mu_{\boldsymbol{\pi}}(s,\boldsymbol{a})=d^{\boldsymbol{\pi}}(s)\boldsymbol{\pi}(\boldsymbol{a}|s)$, where $d^{\boldsymbol{\pi}}(s)=\sum_{t=0}\gamma^t\Pr(s_t=s|\boldsymbol{\pi})$ is the stationary distribution of states given $\boldsymbol{\pi}$. With $\mu_{\boldsymbol{\pi}}$, we can rewrite the optimization objective $J_{\boldsymbol{\rho}}(\boldsymbol{\pi})$ as follows,

$$\begin{aligned}J_{\boldsymbol{\rho}}(\boldsymbol{\pi})&=\sum_{s,\boldsymbol{a}}\mu_{\boldsymbol{\pi}}(s,\boldsymbol{a})r(s,\boldsymbol{a})-\lambda\sum_{s,\boldsymbol{a}}\mu_{\boldsymbol{\pi}}(s,\boldsymbol{a})\log\frac{\boldsymbol{\pi}(\boldsymbol{a}|s)}{\boldsymbol{\rho}(\boldsymbol{a}|s)}\\&=\sum_{s,\boldsymbol{a}}\mu_{\boldsymbol{\pi}}(s,\boldsymbol{a})r(s,\boldsymbol{a})-\lambda D_{\mathrm{C}}\left(\mu_{\boldsymbol{\pi}}\|\mu_{\boldsymbol{\rho}}\right),\end{aligned} \tag{16}$$

where $D_{\mathrm{C}}\left(\mu_{\boldsymbol{\pi}}\|\mu_{\boldsymbol{\rho}}\right)=\sum_{s,\boldsymbol{a}}\mu_{\boldsymbol{\pi}}(s,\boldsymbol{a})\log\frac{\boldsymbol{\pi}(\boldsymbol{a}|s)}{\boldsymbol{\rho}(\boldsymbol{a}|s)}$ is a Bregman divergence (Neu et al., 2017). Therefore, the objective of the divergence regularized MDP can be represented as (17)

$$\boldsymbol{\pi}^*=\arg\max_{\boldsymbol{\pi}}\sum_{s,\boldsymbol{a}}\mu_{\boldsymbol{\pi}}(s,\boldsymbol{a})r(s,\boldsymbol{a})-\lambda D_{\mathrm{C}}\left(\mu_{\boldsymbol{\pi}}\|\mu_{\boldsymbol{\rho}}\right) \tag{17}$$

With this property, similar to Neu et al. (2017) and Wang et al. (2019), we can use the following iterative process,

$$\boldsymbol{\pi}^{t+1} = \arg\max_{\boldsymbol{\pi}} \sum_{s,\boldsymbol{a}} \mu_{\boldsymbol{\pi}}(s,\boldsymbol{a})r(s,\boldsymbol{a}) - \lambda D_{\mathrm{C}}\left(\mu_{\boldsymbol{\pi}}\|\mu_{\boldsymbol{\pi}^t}\right) \tag{18}$$

This iteration is a mirror descent process (Neu et al., 2017), so the convergence of the policy is guaranteed. This process also guarantees that when the policy converges, $D_{\mathrm{C}}\left(\mu_{\boldsymbol{\pi}^{t+1}}\|\mu_{\boldsymbol{\pi}^t}\right) \to 0$; *i.e.*, the regularizer will vanish. Moreover, we can obtain the following inequalities:

$$J(\boldsymbol{\pi}^{t+1}) \geq J(\boldsymbol{\pi}^{t+1}) - \lambda D_{\mathrm{C}}\left(\mu_{\boldsymbol{\pi}^{t+1}}\|\mu_{\boldsymbol{\pi}^t}\right) \geq J(\boldsymbol{\pi}^t) - \lambda D_{\mathrm{C}}\left(\mu_{\boldsymbol{\pi}^t}\|\mu_{\boldsymbol{\pi}^t}\right) = J(\boldsymbol{\pi}^t),$$

The first inequality is from $D_{\mathrm{C}}\left(\mu_{\boldsymbol{\pi}}\|\mu_{\boldsymbol{\rho}}\right) \geq 0$ and the second inequality is from the definition of $\boldsymbol{\pi}^{t+1}$. This conclusion means the policy sequence obtained by this iteration improves monotonically in the original MDP. With these deductions, we actually obtain the following theorem.

**Theorem 2** *By iteratively applying the divergence policy iteration and taking $\boldsymbol{\rho}^k$ as $\boldsymbol{\pi}^k$, the policy sequence $\{\boldsymbol{\pi}^k\}$ will converge and improve monotonically in the original MDP.*

However, it is intractable to perform this update rule in practice because every iteration in (18) needs a convergent policy. Thus, we propose an alternative approximate method. For each agent, we update the policy $\pi_i$ and the target policy $\rho_i$ as $\theta_i = \theta_i + \beta\nabla_{\theta_i}\mathcal{L}_{\boldsymbol{\pi}}$ and $\tilde{\theta}_i = (1-\tau)\tilde{\theta}_i + \tau\theta_i$, where $\beta$ is the learning rate, $\tilde{\theta}_i$ is the weights of $\rho_i$, and $\tau$ is the hyperparameter for soft update. Here we use one gradient step to replace the $\max$ operator in (18). From Theorem 1 and previous discussion, we know that optimizing $\mathcal{L}_{\boldsymbol{\pi}}$ can maximize $J_{\boldsymbol{\rho}}(\boldsymbol{\pi})$, so we use $\nabla_{\theta_i}\mathcal{L}_{\boldsymbol{\pi}}$ in the gradient step for off-policy training instead of the gradient step directly optimizing $J_{\boldsymbol{\rho}}(\boldsymbol{\pi})$ in (16). Moreover, as the convergence of (17) is guaranteed only if the target policy $\boldsymbol{\rho}$ is fixed, we softly update the target policy as the moving average of the policy to prevent the instability caused by the large change of the target policy and hence obtain stable policy improvement.

Now we have all the update rules of DMAC. The training of DMAC is a typical off-policy learning process, which is given in Appendix B for completeness.

## 5 EXPERIMENTS

In this section, we first empirically study the benefits of DMAC and investigate how DMAC improves the performance of existing MARL algorithms in a didactic stochastic game and five SMAC tasks. Then, we demonstrate the advantages of divergence regularizer over entropy regularizer in cooperative MARL.

### 5.1 IMPROVEMENTS OF EXISTING METHODS

DMAC is a flexible framework and can be combined with many existing MARL algorithms. In the experiments, we choose four representative algorithms for different types of methods: COMA (Foerster et al., 2018) for on-policy multi-agent policy gradients, MAAC (Iqbal and Sha, 2019) for off-policy multi-agent actor-critic, QMIX (Rashid et al., 2018) for value decomposition, DOP (Wang et al., 2021b) for the combination of value decomposition and policy gradient. These algorithms need minor modifications to fit the framework of DMAC. We denote these modified algorithms as COMA+DMAC, MAAC+DMAC, QMIX+DMAC, and DOP+DMAC. Generally, our modification is limited and tries to keep the original architecture so as to fairly demonstrate the improvement of DMAC. The details of the modifications are included in Appendix C.1. More details about hyperparameters are available in Appendix C. All the curves in our plots correspond to the mean value of five training runs with different random seeds, and shaded regions indicate 95% confidence interval.

### 5.1.1 A DIDACTIC EXAMPLE

We first test the four groups of methods in a stochastic game where agents share the reward. The stochastic game is generated randomly for the reward function and transition probabilities with 30 states, 3 agents and 5 actions for each agent. Each episode contains 30 steps in this game. The performance of these methods is illustrated in Figure 1. We could find that DMAC performs

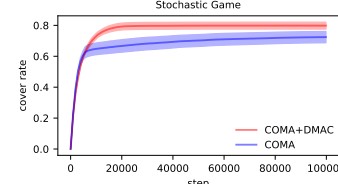

Figure 1: The learning curves in terms of episode rewards of COMA, MAAC, QMIX and DOP groups in the randomly generated stochastic game.

better than the baseline at the end of the training in all the four groups. Moreover, COMA+DMAC, QMIX+DMAC and MAAC+DMAC learn faster than their baselines. Though DOP learns faster than DOP+DMAC at the start, it falls into a sub-optimal policy and DOP+DMAC finds a better policy in the end.

We also show the benefit of exploration in this stochastic game for the convenience of statistics. We evaluate the exploration in terms of the *cover rate* of all state-action pairs, *i.e.*, the ratio of the explored state-action pairs to all state-action pairs. The cover rates of COMA and COMA+DMAC are illustrated in Figure 2. We use COMA here as a representation of the traditional policy gradient method in cooperative MARL. We could find that the cover rate of COMA+DMAC is higher than COMA, which can be an evidence for the benefit of exploration of DMAC. The cover rates of other three groups of algorithms are available in Appendix D.

Figure 2: The learning curves in terms of cover rates of COMA and COMA+DMAC in the randomly generated stochastic game.

### 5.1.2 SMAC

We test all the methods in five tasks of StarCraft Multi-Agent Challenge (SMAC) (Samvelyan et al., 2019). The introduction of the SMAC environment and the training details are included in Appendix C. The learning curves in terms of win rate of all the methods in the five SMAC tasks are illustrated in Figure 3 (four columns for four groups of algorithms and five rows for five different maps in SMAC). In addition, the learning curves in terms of rewards are available in Appendix D. We show the empirical result of DAPO in the map of 3m in the first figure of the second column in Figure 3. It can be seen that DAPO cannot obtain a good performance in the simple task of SMAC, so we skip it in other SMAC tasks. The reason for the low performance of DAPO may be that DAPO omits the correction for $d_\pi(s)/d_{\pi_t}(s)$ in policy update which introduces bias in the gradient of policy, and uses V-trace as off-policy correction which however is biased. These drawbacks may be magnified

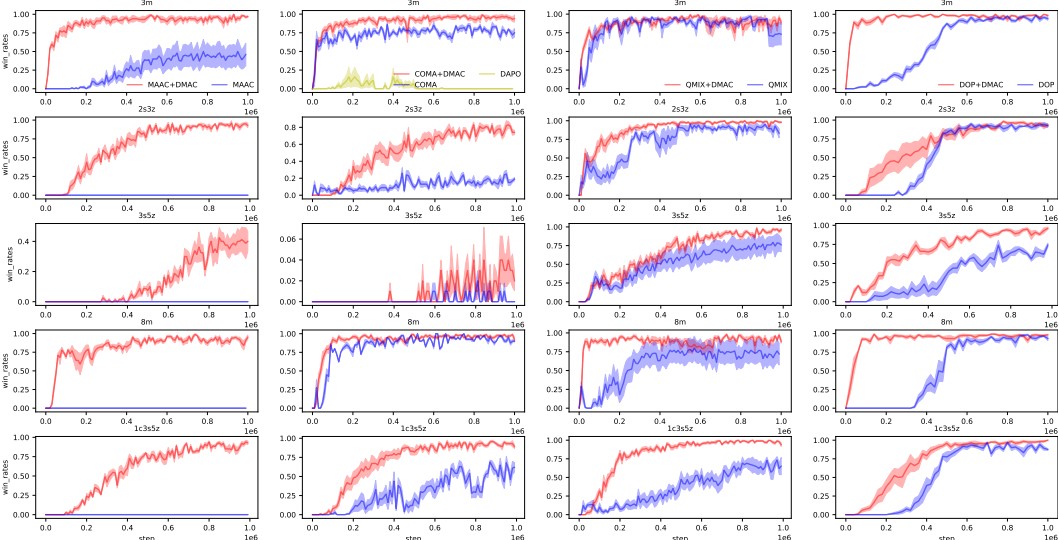

Figure 3: Learning curves in terms of win rates of COMA, MAAC, QMIX and DOP groups in five SMAC maps (each row corresponds to a map and each column corresponds to a group).

Figure 4: The learning curves of win rates of FOP+DMAC and FOP in three SMAC maps.

in MARL settings. The superiority of our naturally off-policy method over the biased off-policy correction method can be partly seen from the large performance gap between COMA+DMAC and DAPO.

In all the five tasks, MAAC+DMAC outperforms MAAC significantly, but MAAC+DMAC does not change the network architecture of MAAC, which shows the benefits of divergence regularizer. As for the result of COMA and COMA+DMAC. We find that COMA+DMAC has higher win rates than COMA in most cases at the end of the training, which can be attributed to the benefits of off-policy training and exploration of divergence regularizer. Though in some cases COMA learns faster than COMA+DMAC, it falls into sub-optimal in the end. This phenomenon can be observed more clearly in the plots of episode rewards in Appendix D, especially in the hard tasks like 3s5z. This can be an evidence for the advantage of divergence regularizer which helps the agents find a better policy.

The stable policy improvement of divergence regularizer can be showed in the variance of the learning curves especially in the comparison between QMIX and QMIX+DMAC. In most tasks, we find that QMIX+DMAC learns substantially faster than QMIX and gets higher win rates in harder tasks. The results of DOP groups are illustrated in the fourth column of Figure 3. DOP+DMAC learns faster than DOP in most cases and finally obtains a better performance. The difference of DOP and DOP+DMAC can also partly show the advantage of naturally off-policy method to the off-policy correction method, as DOP+DMAC replaces the tree backup loss with off-policy TD($\lambda$).

DMAC improves the performance and/or convergence speed of the evaluated algorithms in most tasks. This empirically demonstrates the benefits of divergence regularizer. Moreover, the superiority of our naturally off-policy learning over the biased off-policy correction method can be partly witnessed from the empirical results.

## 5.2 Comparison with Entropy Regularization

FOP (Zhang et al., 2021) combines the value decomposition with entropy regularization, which obtained the state-of-the-art performance in SMAC tasks. FOP has a well tuned scheme for the temperature parameter of the entropy, so we take FOP as a strong baseline for entropy-regularized methods in cooperative MARL. We compare FOP and FOP+DMAC in three SMAC tasks, 3s_vs_3z, 2c_vs_64zg, and MMM2, which respectively correspond to the three levels of difficulties (*i.e.*, easy, hard, and super hard) for SMAC tasks. These tasks are taken from the original paper of FOP. The modifications of FOP+DMAC are also included in Appendix C.1. The win rates of FOP and FOP+DMAC are illustrated in Figure 4. We could find that FOP+DMAC learns much faster than FOP in 3s_vs_3z, while it performs better than FOP in other two harder tasks. These results could be an evidence for the advantages of DMAC and the bias of entropy regularization.

## 6 Conclusion

We propose a multi-agent actor-critic framework, DMAC, for cooperative MARL. We investigate divergence regularization, derive divergence policy iteration, and deduce the update rules for the critic, policy, and target policy in multi-agent settings. DMAC is a naturally off-policy framework and the divergence regularizer is beneficial to exploration and stable policy improvement. Unlike entropy regularizer, the divergence regularizer will not bias the converged policy. DMAC is also a flexible framework and can be combined with many existing MARL algorithms with limited modification. It is empirically demonstrated that combining DMAC with existing MARL algorithms can improve the performance and convergence speed in a stochastic game and SMAC tasks.

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

# A  PROOFS

## A.1  LEMMA 1

***Proof.***   We define a new reward function

$$r_{\boldsymbol{\rho}}^{\boldsymbol{\pi}}(s, \boldsymbol{a}) = r(s, \boldsymbol{a}) - \lambda \mathbb{E}_{s' \sim P(\cdot|s, \boldsymbol{a})} \left[ D_{\mathrm{KL}} \left( \boldsymbol{\pi}(\cdot|s') \| \boldsymbol{\rho}(\cdot|s') \right) \right],$$

then we can rewrite the definition of operator $\Gamma_{\boldsymbol{\rho}}^{\boldsymbol{\pi}}$ as

$$\Gamma_{\boldsymbol{\rho}}^{\boldsymbol{\pi}} Q(s, \boldsymbol{a}) = r_{\boldsymbol{\rho}}^{\boldsymbol{\pi}}(s, \boldsymbol{a}) + \gamma \mathbb{E}_{s' \sim P(\cdot|s, \boldsymbol{a}), \boldsymbol{a}' \sim \boldsymbol{\pi}(\cdot|s')} \left[ Q(s', \boldsymbol{a}') \right].$$

With this formula, we can apply the traditional convergence result of policy evaluation in Sutton and Barto (2018).  □

## A.2  LEMMA 2

For the proof of Lemma 2, we need the following lemma (Haarnoja et al., 2018) about improving policy in entropy-regularized MDP.

**Lemma 4**  *If we have a new policy $\boldsymbol{\pi}_{\mathrm{new}}$ and*

$$\boldsymbol{\pi}_{\mathrm{new}} = \arg \min_{\boldsymbol{\pi}} D_{\mathrm{KL}} \left( \boldsymbol{\pi}(\cdot|s) \| \frac{\exp \left( Q_{\mathrm{ent}}^{\boldsymbol{\pi}_{\mathrm{old}}}(s, \cdot)/\lambda \right)}{Z^{\boldsymbol{\pi}_{\mathrm{old}}}(s)} \right),$$

*where $Z^{\boldsymbol{\pi}_{\mathrm{old}}}(s)$ represents the normalization term, then we have*

$$Q_{\mathrm{ent}}^{\boldsymbol{\pi}_{\mathrm{new}}}(s, a) \geq Q_{\mathrm{ent}}^{\boldsymbol{\pi}_{\mathrm{old}}}(s, a), \quad \forall s \in S, a \in A.$$

With Lemma 4, we have the following proof of Lemma 2.

***Proof.***   Let $\hat{Q}$ be the same as the definition in Proof A.5. Then we have $\hat{Q}^{\boldsymbol{\pi}}(s, \boldsymbol{a}) = Q_{\boldsymbol{\rho}}^{\boldsymbol{\pi}}(s, \boldsymbol{a}) + \lambda \log \boldsymbol{\rho}(\boldsymbol{a}|s), \forall \boldsymbol{\pi}$.

According to Lemma 4,

$$\hat{\boldsymbol{\pi}}_{\mathrm{new}}(s, \cdot) = \arg \min_{\boldsymbol{\pi}} D_{\mathrm{KL}} \left( \boldsymbol{\pi}(\cdot|s) \| \frac{\exp \left( \hat{Q}^{\boldsymbol{\pi}_{\mathrm{old}}}(s, \cdot)/\lambda \right)}{Z^{\boldsymbol{\pi}_{\mathrm{old}}}(s)} \right)$$

$$\hat{Q}^{\hat{\boldsymbol{\pi}}_{\mathrm{new}}}(s, \boldsymbol{a}) \geq \hat{Q}^{\boldsymbol{\pi}_{\mathrm{old}}}(s, \boldsymbol{a}), \quad \forall \boldsymbol{a} \in A.$$

With the definition, we have

$$D_{\mathrm{KL}} \left( \boldsymbol{\pi}(\cdot|s) \| \frac{\exp(\hat{Q}^{\boldsymbol{\pi}_{\mathrm{old}}}(s, \cdot)/\lambda)}{Z^{\boldsymbol{\pi}_{\mathrm{old}}}(s)} \right) = D_{\mathrm{KL}} \left( \boldsymbol{\pi}(\cdot|s) \| \boldsymbol{\rho}(\cdot|s) \frac{\exp(Q_{\boldsymbol{\rho}}^{\boldsymbol{\pi}_{\mathrm{old}}}(s, \cdot)/\lambda)}{Z^{\boldsymbol{\pi}_{\mathrm{old}}}(s)} \right)$$

$$\boldsymbol{\pi}_{\mathrm{new}} = \hat{\boldsymbol{\pi}}_{\mathrm{new}}$$

$$Q_{\boldsymbol{\rho}}^{\boldsymbol{\pi}_{\mathrm{new}}}(s, \boldsymbol{a}) = \hat{Q}^{\boldsymbol{\pi}_{\mathrm{new}}}(s, \boldsymbol{a}) - \lambda \log \boldsymbol{\rho}(\boldsymbol{a}|s) \geq \hat{Q}^{\boldsymbol{\pi}_{\mathrm{old}}}(s, \boldsymbol{a}) - \lambda \log \boldsymbol{\rho}(\boldsymbol{a}|s) = Q_{\boldsymbol{\rho}}^{\boldsymbol{\pi}_{\mathrm{old}}}(s, \boldsymbol{a}).$$

□

## A.3  LEMMA 3

***Proof.***   From the equation

$$\pi_{\mathrm{new}}^i = \arg \max_{\pi_i} \sum_{a_i} \pi_i(a_i|s) \left( k_i(s) Q_{\boldsymbol{\rho}}^{\boldsymbol{\pi}_{\mathrm{old}}^i}(s, a_i) - \lambda \log \frac{\pi_i(a_i|s)}{\rho_i(a_i|s)} \right),$$

we can obtain

$$\sum_{a_i} \pi_{\mathrm{new}}^i(a_i|s) \left( k_i(s) Q_{\boldsymbol{\rho}}^{\boldsymbol{\pi}_{\mathrm{old}}^i}(s, a_i) - \lambda \log \frac{\pi_{\mathrm{new}}^i(a_i|s)}{\rho_i(a_i|s)} \right)$$

$$\geq \sum_{a_i} \pi_{\mathrm{old}}^i(a_i|s) \left( k_i(s) Q_{\boldsymbol{\rho}}^{\boldsymbol{\pi}_{\mathrm{old}}^i}(s, a_i) - \lambda \log \frac{\pi_{\mathrm{old}}^i(a_i|s)}{\rho_i(a_i|s)} \right). \tag{19}$$

By taking expectation on the both side of (19), we can obtain the followings.

$$
\sum_{a_{-i}} \tilde{\pi}_{-i}(a_{-i}|s) \sum_{a_i} \pi^i_{\text{new}}(a_i|s) \left( k_i(s) Q^{\pi^i_{\text{old}}}_\rho(s, a_i) - \lambda \log \frac{\pi^i_{\text{new}}(a_i|s)}{\rho_i(a_i|s)} \right)
$$
$$
\geq \sum_{a_{-i}} \tilde{\pi}_{-i}(a_{-i}|s) \sum_{a_i} \pi^i_{\text{old}}(a_i|s) \left( k_i(s) Q^{\pi^i_{\text{old}}}_\rho(s, a_i) - \lambda \log \frac{\pi^i_{\text{old}}(a_i|s)}{\rho_i(a_i|s)} \right) \tag{20}
$$
$$
\forall i \in I \quad \forall \tilde{\pi}_{-i}
$$

Moreover, we can easily have the following derivation,

$$
\sum_{a_i} \pi^i_{\text{new}}(a_i|s) \left( k_i(s) Q^{\pi^i_{\text{old}}}_\rho(s, a_i) + b(s) - \lambda \log \frac{\pi^i_{\text{new}}(a_i|s)}{\rho_i(a_i|s)} \right)
$$
$$
= \sum_{a_{-i}} u_1(a_{-i}|s) \sum_{a_i} \pi^i_{\text{new}}(a_i|s) \left( k_i(s) Q^{\pi^i_{\text{old}}}_\rho(s, a_i) + b(s) - \lambda \log \frac{\pi^i_{\text{new}}(a_i|s)}{\rho_i(a_i|s)} \right)
$$
$$
= \sum_{a_{-i}} u_2(a_{-i}|s) \sum_{a_i} \pi^i_{\text{new}}(a_i|s) \left( k_i(s) Q^{\pi^i_{\text{old}}}_\rho(s, a_i) + b(s) - \lambda \log \frac{\pi^i_{\text{new}}(a_i|s)}{\rho_i(a_i|s)} \right) \tag{21}
$$
$$
\forall u_1, u_2.
$$

Then we have

$$
\mathbb{E}_{\boldsymbol{a} \sim \boldsymbol{\pi}_{\text{new}}} \left[ Q^{\boldsymbol{\pi}_{\text{old}}}_\rho(s, \boldsymbol{a}) - \log \frac{\boldsymbol{\pi}_{\text{new}}(\boldsymbol{a}|s)}{\boldsymbol{\rho}(\boldsymbol{a}|s)} \right]
$$
$$
= \sum_{\boldsymbol{a}} \boldsymbol{\pi}_{\text{new}}(\boldsymbol{a}|s) \sum_i \left( k_i(s) Q^{\pi^i_{\text{old}}}_\rho(s, a_i) + b(s) - \lambda \log \frac{\pi^i_{\text{new}}(a_i|s)}{\rho_i(a_i|s)} \right)
$$
$$
= \sum_{\boldsymbol{a}} \boldsymbol{\pi}_{\text{new}}(\boldsymbol{a}|s) \sum_i \left( k_i(s) Q^{\pi^i_{\text{old}}}_\rho(s, a_i) + b(s) - \lambda \log \frac{\pi^i_{\text{new}}(a_i|s)}{\rho_i(a_i|s)} \right)
$$
$$
= \sum_i \sum_{a_{-i}} \pi^{-i}_{\text{new}}(a_{-i}|s) \sum_{a_i} \pi^i_{\text{new}}(a_i|s) \left( k_i(s) Q^{\pi^i_{\text{old}}}_\rho(s, a_i) + b(s) - \lambda \log \frac{\pi^i_{\text{new}}(a_i|s)}{\rho_i(a_i|s)} \right) \tag{22}
$$
$$
= \sum_i \sum_{a_{-i}} \pi^{-i}_{\text{old}}(a_{-i}|s) \sum_{a_i} \pi^i_{\text{new}}(a_i|s) \left( k_i(s) Q^{\pi^i_{\text{old}}}_\rho(s, a_i) + b(s) - \lambda \log \frac{\pi^i_{\text{new}}(a_i|s)}{\rho_i(a_i|s)} \right)
$$
$$
\geq \sum_i \sum_{a_{-i}} \pi^{-i}_{\text{old}}(a_{-i}|s) \sum_{a_i} \pi^i_{\text{old}}(a_i|s) \left( k_i(s) Q^{\pi^i_{\text{old}}}_\rho(s, a_i) + b(s) - \lambda \log \frac{\pi^i_{\text{old}}(a_i|s)}{\rho_i(a_i|s)} \right)
$$
$$
= V^{\boldsymbol{\pi}_{\text{old}}}_\rho(s)
$$

The fourth equation is from (21) and the fifth inequality is from (20).

By repeatedly applying (22) and the relation $Q^{\boldsymbol{\pi}_{\text{old}}}_\rho(s, \boldsymbol{a}) = r(s, \boldsymbol{a}) + \gamma \mathbb{E}_{s'} \left[ V^{\boldsymbol{\pi}_{\text{old}}}_\rho(s') \right]$, we can complete the proof as followings.

$$
Q^{\boldsymbol{\pi}_{\text{old}}}_\rho(s, \boldsymbol{a}) = r(s, \boldsymbol{a}) + \gamma \mathbb{E}_{s'} \left[ V^{\boldsymbol{\pi}_{\text{old}}}_\rho(s') \right]
$$
$$
\leq r(s, \boldsymbol{a}) + \gamma \mathbb{E}_{s'} \left[ \mathbb{E}_{\boldsymbol{a}' \sim \boldsymbol{\pi}_{\text{new}}} \left[ Q^{\boldsymbol{\pi}_{\text{old}}}_\rho(s', , \boldsymbol{a}') - \log \frac{\boldsymbol{\pi}_{\text{new}}(\boldsymbol{a}'|s')}{\boldsymbol{\rho}(\boldsymbol{a}'|s')} \right] \right]
$$
$$
= r(s, \boldsymbol{a}) + \gamma \mathbb{E}_{s'} \left[ \mathbb{E}_{\boldsymbol{a}' \sim \boldsymbol{\pi}_{\text{new}}} \left[ r(s', \boldsymbol{a}') + \gamma \mathbb{E}_{s''} \left[ V^{\boldsymbol{\pi}_{\text{old}}}_\rho(s'') \right] - \log \frac{\boldsymbol{\pi}_{\text{new}}(\boldsymbol{a}'|s')}{\boldsymbol{\rho}(\boldsymbol{a}'|s')} \right] \right]
$$
$$
\cdots
$$
$$
\leq Q^{\boldsymbol{\pi}_{\text{new}}}_\rho(s, \boldsymbol{a})
$$

$\square$

### A.4 THEOREM 1

***Proof.*** First, we will show that Divergence Policy Iteration will monotonically improve the policy. From Lemma 2, we know that

$$
\begin{aligned}
V_{\boldsymbol{\rho}}^{\boldsymbol{\pi}_{\text{new}}}(s) &= \mathbb{E}_{\boldsymbol{a}\sim\boldsymbol{\pi}_{\text{new}}(\cdot|s)}\left[Q_{\boldsymbol{\rho}}^{\boldsymbol{\pi}_{\text{new}}}(s,\boldsymbol{a}) - \lambda\log\frac{\boldsymbol{\pi}_{\text{new}}(\boldsymbol{a}|s)}{\boldsymbol{\rho}(\boldsymbol{a}|s)}\right] \\
&\geq \mathbb{E}_{\boldsymbol{a}\sim\boldsymbol{\pi}_{\text{new}}(\cdot|s)}\left[Q_{\boldsymbol{\rho}}^{\boldsymbol{\pi}_{\text{old}}}(s,\boldsymbol{a}) - \lambda\log\frac{\boldsymbol{\pi}_{\text{new}}(\boldsymbol{a}|s)}{\boldsymbol{\rho}(\boldsymbol{a}|s)}\right] \\
&\geq \mathbb{E}_{\boldsymbol{a}\sim\boldsymbol{\pi}_{\text{old}}(\cdot|s)}\left[Q_{\boldsymbol{\rho}}^{\boldsymbol{\pi}_{\text{old}}}(s,\boldsymbol{a}) - \lambda\log\frac{\boldsymbol{\pi}_{\text{old}}(\boldsymbol{a}|s)}{\boldsymbol{\rho}(\boldsymbol{a}|s)}\right] \\
&= V_{\boldsymbol{\rho}}^{\boldsymbol{\pi}_{\text{old}}}(s).
\end{aligned}
$$

The first inequality is from the conclusion of Lemma 2 that

$$
Q_{\boldsymbol{\rho}}^{\boldsymbol{\pi}_{\text{new}}}(s,\boldsymbol{a}) \geq Q_{\boldsymbol{\rho}}^{\boldsymbol{\pi}_{\text{old}}}(s,\boldsymbol{a}) \quad \forall \boldsymbol{a} \in A,
$$

and the second inequality is from the definition of $\boldsymbol{\pi}_{\text{new}}$ that

$$
\boldsymbol{\pi}_{\text{new}} = \arg\min_{\boldsymbol{\pi}} D_{\text{KL}}\left(\boldsymbol{\pi}(\cdot|s)\|\frac{\exp\left(Q_{\boldsymbol{\rho}}^{\boldsymbol{\pi}_{\text{old}}}(s,\cdot)/\lambda\right)}{Z^{\boldsymbol{\pi}_{\text{old}}}(s)}\right).
$$

Here we have $V^{\boldsymbol{\pi}_{\text{new}}}(s) \geq V^{\boldsymbol{\pi}_{\text{old}}}(s)$, $\forall s \in S$, and thus $J_{\boldsymbol{\rho}}(\boldsymbol{\pi}_{\text{new}}) \geq J_{\boldsymbol{\rho}}(\boldsymbol{\pi}_{\text{old}})$. So, Divergence Policy Iteration will monotonically improve the policy.

Since the $Q_{\boldsymbol{\rho}}^{\boldsymbol{\pi}}$ is bounded above (the reward function is bounded), the sequence of Q-function $\{Q^k\}$ of Divergence Policy Iteration will converge and the corresponding policy sequence will also converge to some policy $\boldsymbol{\pi}_{\text{conv}}$. We need to show $\boldsymbol{\pi}_{\text{conv}} = \boldsymbol{\pi}_{\boldsymbol{\rho}}^*$.

$$
\begin{aligned}
V_{\boldsymbol{\rho}}^{\boldsymbol{\pi}_{\text{conv}}}(s) &= \mathbb{E}_{\boldsymbol{a}\sim\boldsymbol{\pi}_{\text{conv}}(\cdot|s)}\left[Q_{\boldsymbol{\rho}}^{\boldsymbol{\pi}_{\text{conv}}}(s,\boldsymbol{a}) - \lambda\log\frac{\boldsymbol{\pi}_{\text{conv}}(\boldsymbol{a}|s)}{\boldsymbol{\rho}(\boldsymbol{a}|s)}\right] \\
&\geq \mathbb{E}_{\boldsymbol{a}\sim\boldsymbol{\pi}(\cdot|s)}\left[Q_{\boldsymbol{\rho}}^{\boldsymbol{\pi}_{\text{conv}}}(s,\boldsymbol{a}) - \lambda\log\frac{\boldsymbol{\pi}(\boldsymbol{a}|s)}{\boldsymbol{\rho}(\boldsymbol{a}|s)}\right] \\
&\geq \mathbb{E}_{\boldsymbol{a}\sim\boldsymbol{\pi}(\cdot|s)}\left[r(s,\boldsymbol{a}) + \gamma\mathbb{E}_{\boldsymbol{a}'\sim\boldsymbol{\pi}(\cdot|s')}\left[Q_{\boldsymbol{\rho}}^{\boldsymbol{\pi}_{\text{conv}}}(s',\boldsymbol{a}') - \lambda\log\frac{\boldsymbol{\pi}(\boldsymbol{a}'|s')}{\boldsymbol{\rho}(\boldsymbol{a}'|s')}\right] - \lambda\log\frac{\boldsymbol{\pi}(\boldsymbol{a}|s)}{\boldsymbol{\rho}(\boldsymbol{a}|s)}\right] \\
&\cdots \\
&\geq V_{\boldsymbol{\rho}}^{\boldsymbol{\pi}}(s).
\end{aligned}
$$

The first inequality is from the definition of $\boldsymbol{\pi}_{\text{conv}}$ that

$$
\boldsymbol{\pi}_{\text{conv}} = \arg\min_{\boldsymbol{\pi}} D_{\text{KL}}\left(\boldsymbol{\pi}(\cdot|s)\|\frac{\exp\left(Q_{\boldsymbol{\rho}}^{\boldsymbol{\pi}_{\text{conv}}}(s,\cdot)/\lambda\right)}{Z^{\boldsymbol{\pi}_{\text{conv}}}(s)}\right)
$$

and all the other inequalities are just iteratively using the first inequality and the relation of $Q_{\boldsymbol{\rho}}^{\boldsymbol{\pi}}$ and $V_{\boldsymbol{\rho}}^{\boldsymbol{\pi}}$. With iterations, we replace all the $\boldsymbol{\pi}_{\text{conv}}$ with $\boldsymbol{\pi}$ in the expression of $V_{\boldsymbol{\rho}}^{\boldsymbol{\pi}_{\text{conv}}}(s)$ and finally we get $V_{\boldsymbol{\rho}}^{\boldsymbol{\pi}}(s)$. Therefore, we have

$$
\begin{aligned}
V_{\boldsymbol{\rho}}^{\boldsymbol{\pi}_{\text{conv}}}(s) &\geq V_{\boldsymbol{\rho}}^{\boldsymbol{\pi}}(s) \quad \forall s \in S \quad \forall \boldsymbol{\pi} \in \Pi \\
J_{\boldsymbol{\rho}}(\boldsymbol{\pi}_{\text{conv}}) &\geq J_{\boldsymbol{\rho}}(\boldsymbol{\pi}) \quad \forall \boldsymbol{\pi} \in \Pi \\
\boldsymbol{\pi}_{\text{conv}} &= \boldsymbol{\pi}_{\boldsymbol{\rho}}^*.
\end{aligned}
$$

$\square$

### A.5 PROPOSITION 1

Before the proof of Proposition 1, we need some results about the optimal Q-function $Q_{\text{ent}}^*$, the optimal V-function $V_{\text{ent}}^*$, and the optimal policy $\pi_{\text{ent}}^*$ in entropy-regularized MDP. We have the following lemma (Nachum et al., 2017).

**Lemma 5**

$$\boldsymbol{\pi}_{\text{ent}}^*(s,a) \propto \exp\left(\left(r(s,a) + \gamma\mathbb{E}_{s'\sim P(\cdot|s,a)}\left[V_{\text{ent}}^*(s')\right]\right)/\lambda\right)$$

$$V_{\text{ent}}^*(s) = \lambda\log\sum_a \exp\left(\left(r(s,a) + \gamma\mathbb{E}_{s'\sim P(\cdot|s,a)}\left[V_{\text{ent}}^*(s')\right]\right)/\lambda\right)$$

$$Q_{\text{ent}}^*(s,a) = r(s,a) + \gamma\lambda\mathbb{E}_{s'\sim P(\cdot|s,a)}\left[\log\sum_{a'}\exp\left(Q_{\text{ent}}^*(s',a')/\lambda\right)\right]$$

With Lemma 5, we can complete the proof of Proposition 1.

***Proof.*** Let $\hat{r}(s,\boldsymbol{a}) = r(s,\boldsymbol{a}) + \lambda\log\boldsymbol{\rho}(\boldsymbol{a}|s)$, we consider the objective function

$$\hat{J}(\boldsymbol{\pi}) = \mathbb{E}_{\boldsymbol{\pi}}\left[\sum_{t=0}\gamma^t\left(\hat{r}(s_t,\boldsymbol{a}_t) - \lambda\log\boldsymbol{\pi}(\boldsymbol{a}_t|s_t)\right)\right].$$

Let $\hat{\boldsymbol{\pi}}^*(\boldsymbol{a}|s), \hat{V}^*(s)$ and $\hat{Q}^*(s,\boldsymbol{a})$ be the corresponding optimal policy, V-function and Q-function of $\hat{J}(\boldsymbol{\pi})$. By definition we can obtain

$$\hat{\boldsymbol{\pi}}^*(\boldsymbol{a}|s) = \boldsymbol{\pi}_{\boldsymbol{\rho}}^*(\boldsymbol{a}|s)$$

$$\hat{V}^*(s) = \mathbb{E}_{\boldsymbol{a}\sim\hat{\boldsymbol{\pi}}^*(\cdot|s),s'\sim P(\cdot|s,a)}\left[\hat{r}(s,\boldsymbol{a}) + \gamma\hat{V}^*(s') - \lambda\log\hat{\boldsymbol{\pi}}^*(\boldsymbol{a}|s)\right]$$

$$= \mathbb{E}_{\boldsymbol{a}\sim\boldsymbol{\pi}_{\boldsymbol{\rho}}^*(\cdot|s),s'\sim P(\cdot|s,a)}\left[r(s,\boldsymbol{a}) + \gamma\hat{V}^*(s') - \lambda\log\frac{\boldsymbol{\pi}_{\boldsymbol{\rho}}^*(\boldsymbol{a}|s)}{\boldsymbol{\rho}(\boldsymbol{a}|s)}\right]$$

$$= V_{\boldsymbol{\rho}}^*(s)$$

$$\hat{Q}^*(s,\boldsymbol{a}) = \hat{r}(s,\boldsymbol{a}) + \mathbb{E}_{s'\sim P(\cdot|s,a)}\left[\hat{V}^*(s')\right]$$

$$= r(s,\boldsymbol{a}) + \mathbb{E}_{s'\sim P(\cdot|s,a)}\left[V_{\boldsymbol{\rho}}^*(s')\right] + \lambda\log\boldsymbol{\rho}(\boldsymbol{a}|s)$$

$$= Q_{\boldsymbol{\rho}}^*(s,\boldsymbol{a}) + \lambda\log\boldsymbol{\rho}(\boldsymbol{a}|s).$$

According to Lemma 5, we have

$$\boldsymbol{\pi}_{\boldsymbol{\rho}}^*(\boldsymbol{a}|s) = \hat{\boldsymbol{\pi}}^*(\boldsymbol{a}|s) \propto \exp\left(\left(\hat{r}(s,\boldsymbol{a}) + \gamma\mathbb{E}_{s'\sim P(\cdot|s,a)}\left[\hat{V}^*(s')\right]\right)/\lambda\right)$$

$$= \boldsymbol{\rho}(\boldsymbol{a}|s)\exp\left(\left(r(s,\boldsymbol{a}) + \gamma\mathbb{E}_{s'\sim P(\cdot|s,a)}\left[V_{\boldsymbol{\rho}}^*(s')\right]\right)/\lambda\right)$$

$$V_{\boldsymbol{\rho}}^*(s) = \hat{V}^*(s)$$

$$= \lambda\log\sum_a \exp\left(\left(\hat{r}(s,\boldsymbol{a}) + \gamma\mathbb{E}_{s'\sim P(\cdot|s,a)}\left[\hat{V}^*(s')\right]\right)/\lambda\right)$$

$$= \lambda\log\sum_a \boldsymbol{\rho}(\boldsymbol{a}|s)\exp\left(\left(r(s,\boldsymbol{a}) + \gamma\mathbb{E}_{s'\sim P(\cdot|s,a)}\left[V_{\boldsymbol{\rho}}^*(s')\right]\right)/\lambda\right)$$

$$Q_{\boldsymbol{\rho}}^*(s,\boldsymbol{a}) = \hat{Q}^*(s,\boldsymbol{a}) - \lambda\log\boldsymbol{\rho}(\boldsymbol{a}|s)$$

$$= \hat{r}(s,\boldsymbol{a}) + \gamma\lambda\mathbb{E}_{s'\sim P(\cdot|s,a)}\left[\log\sum_{a'}\exp\left(\hat{Q}^*(s',\boldsymbol{a}')/\lambda\right)\right] - \lambda\log\boldsymbol{\rho}(\boldsymbol{a}|s)$$

$$= r(s,\boldsymbol{a}) + \gamma\lambda\mathbb{E}_{s'\sim P(\cdot|s,a)}\left[\log\sum_{a'}\boldsymbol{\rho}(\boldsymbol{a}|s)\exp\left(Q_{\boldsymbol{\rho}}^*(s',\boldsymbol{a}')/\lambda\right)\right].$$

$\square$

### A.6 Derivation of Gradient

$$\nabla_{\theta_i}\mathcal{L}_{\boldsymbol{\pi}} = \mathbb{E}_{s\sim\mathcal{D}}\left[\sum_{\boldsymbol{a}}\nabla_{\theta_i}\boldsymbol{\pi}(\boldsymbol{a}|s)\left(Q_{\boldsymbol{\rho}}^{\boldsymbol{\pi}}(s,\boldsymbol{a}) - \lambda\log\frac{\boldsymbol{\pi}(\boldsymbol{a}|s)}{\boldsymbol{\rho}(\boldsymbol{a}|s)}\right) + \boldsymbol{\pi}(\boldsymbol{a}|s)\nabla_{\theta_i}(-\lambda\log\frac{\boldsymbol{\pi}(\boldsymbol{a}|s)}{\boldsymbol{\rho}(\boldsymbol{a}|s)})\right]$$

$$= \mathbb{E}_{s\sim\mathcal{D}}\left[\sum_{\boldsymbol{a}}\boldsymbol{\pi}(\boldsymbol{a}|s)\nabla_{\theta_i}\log\pi_i(a_i|s)\left(Q_{\boldsymbol{\rho}}^{\boldsymbol{\pi}}(s,\boldsymbol{a}) - \lambda\log\frac{\boldsymbol{\pi}(\boldsymbol{a}|s)}{\boldsymbol{\rho}(\boldsymbol{a}|s)}\right) - \lambda\boldsymbol{\pi}(\boldsymbol{a}|s)\nabla_{\theta_i}\log\pi_i(a_i|s)\right]$$

$$= \mathbb{E}_{s\sim\mathcal{D},a\sim\boldsymbol{\pi}}\left[\nabla_{\theta_i}\log\pi_i(a_i|s)\left(Q_{\boldsymbol{\rho}}^{\boldsymbol{\pi}}(s,\boldsymbol{a}) - \lambda\log\frac{\boldsymbol{\pi}(\boldsymbol{a}|s)}{\boldsymbol{\rho}(\boldsymbol{a}|s)} - \lambda\right)\right]$$

## B  Algorithm

Algorithm 1 gives the training procedure of DMAC.

---
**Algorithm 1** DMAC
---
1: **for** episode = 1 to $m$ **do**
2:     Initialize the environment and receive initial state $s$
3:     **for** $t = 1$ to max-episode-length **do**
4:         For each agent $i$, select action $a_i \sim \pi_i(\cdot|s)$
5:         Execute joint-action $\boldsymbol{a} = (a_1, a_2, \cdots, a_n)$ and observe reward $r$ and next state $s'$
6:         Store $(s, \boldsymbol{a}, r, s')$ in replay buffer $\mathcal{D}$
7:     **end for**
8:     Sample a random minibatch of $K$ samples from $\mathcal{D}$, $\{(s_k, \boldsymbol{a}_k, r_k, s'_k)\}_K$
9:     **for** agent $i = 1$ to $n$ **do**
10:         Update policy $\pi_i$: $\theta_i = \theta_i + \beta\nabla_{\theta_i}\mathcal{L}_{\boldsymbol{\pi}}$
11:         Update target policy $\rho_i$: $\tilde{\theta}_i = (1-\tau)\tilde{\theta}_i + \tau\theta_i$
12:     **end for**
13:     Update critic: $\phi = \phi - \alpha\nabla_{\phi}\mathcal{L}_Q$
14:     Update target critic: $\tilde{\phi} = (1-\tau)\tilde{\phi} + \tau\phi$
15: **end for**
---

## C  Implementation Details

SMAC is a MARL environment based on the game StarCraft II (SC2). Agents control different units in SC2 and can attack, move or take other actions. The general mode of SMAC tasks is that agents control a team of units to counter another team controlled by built-in AI. The target of agents is to wipe out all the enemy units and agents will gain rewards when hurting or killing enemy units. Agents have an observation range and can only observe information of units in this range, but the information of all the units can be accessed in training. We test all the methods in totally 8 tasks/maps: 3m, 2s3z, 3s5z, 8m,1c3s5z,3s_vs_3z,2c_vs_64zg, and MMM2.

In SMAC tasks, we evaluate 20 episodes in every 10000 training steps in the one million steps training procedure for COMA, MAAC, QMIX and in every 20000 time steps for DOP and FOP. In evaluation, we select greedy actions for QMIX and FOP (following the setting in the FOP paper) and sample actions according to action distribution for stochastic policy (COMA, MAAC, DOP and divergence-regularized methods).

### C.1  Modifications of the Baseline Methods

The modifications of the baseline methods, COMA,MAAC,QMIX,DOP and FOP, are as follows.

- COMA. We keep the original critic and actor networks and add a target policy network with the same architecture as the actor. As COMA is on-policy but COMA+DMAC is off-policy, we add a replay buffer for experience replay.

- MAAC already has a target policy for stability, so we do not need to modify the network architecture. We only change the update rule for the critic and actors.

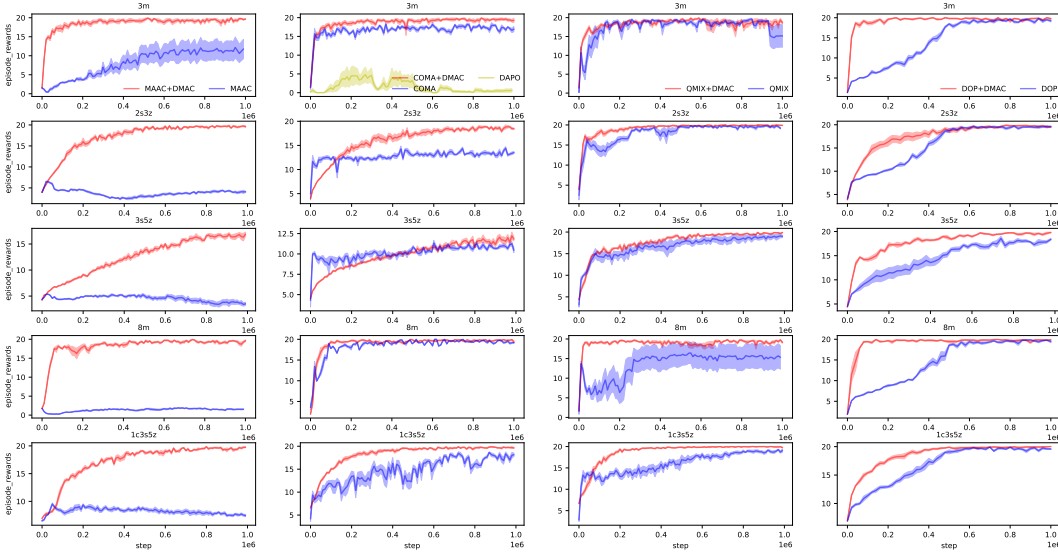

Figure 5: Learning curves in terms of cover rates of COMA, QMIX, MAAC and DOP groups in the randomly generated stochastic game.

- **QMIX** is a value-based method, so we need to add a policy network and a target policy network for each agent. We keep the original individual Q-functions to learn the critic in QMIX+DMAC. In divergence-regularized MDP, the max operator is not needed in the critic update, so we abandon the hypernet and use an MLP, which takes individual Q-values and state as input and produces the joint Q-value. This architecture is simple and its expressive capability is not limited by QMIX's IGM condition.

- **DOP.** We keep the original critic and actor networks and add a target policy network with the same architecture as the actor. We keep the value decomposition structure and use off-policy TD($\lambda$) for all samples in training to replace the tree backup loss and on-policy TD($\lambda$) loss.

- **FOP.** We replace the entropy regularizers with divergence regularizers in FOP and use the update rules of DMAC. We keep the original architecture of FOP.

### C.2 HYPERPARAMETERS

As all tasks in our experiments are cooperative with shared reward, so we use parameter-sharing policy network and critic network for MAAC and MAAC+DMAC to accelerate training. Besides, we add a RNN layer to the policy network and critic network in MAAC and MAAC+DMAC to settle the partial observability.

All the policy networks are the same as two linear layers and one GRUCell layer with ReLU activation and the number of hidden units is 64. The individual Q-networks for QMIX group is the same as the policy network mentioned before. The critic network for COMA group is a MLP with three 128-unit hidden layers and ReLU activation. The attention dimension in the critic networks of MAAC group is 32. The number of hidden units of mixer network in QMIX group is 32. The learning rate for critic is $10^{-3}$ and the learning rate for actor is $10^{-4}$. We train all networks with RMSprop optimizer. The discouted factor is $\gamma = 0.99$. The coefficient of regularizer is $\lambda = 0.01$. The *td_lambda* factor used in COMA group is $0.8$. The parameter used for soft updating target policy is $\tau = 0.01$. Our

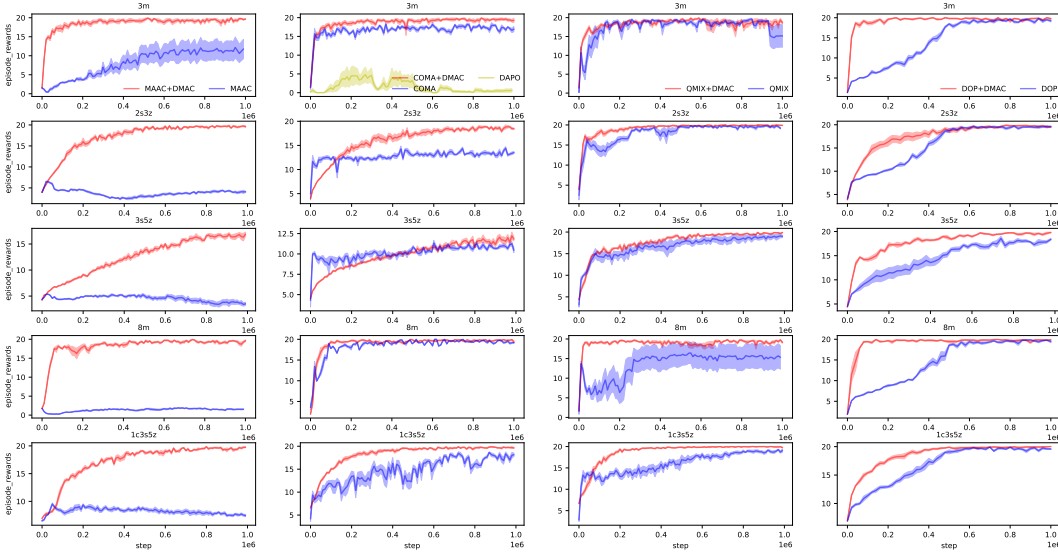

Figure 6: Learning curves in terms of mean episode reward of COMA, MAAC, QMIX, and DOP groups in five SMAC maps (each row corresponds to a map and each column corresponds to a group).

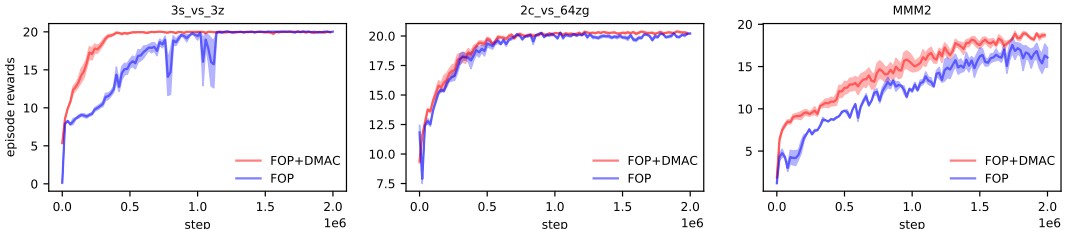

Figure 7: Learning curves in terms of mean episode rewards of FOP+DMAC and FOP in three SMAC maps (each column corresponds to a map).

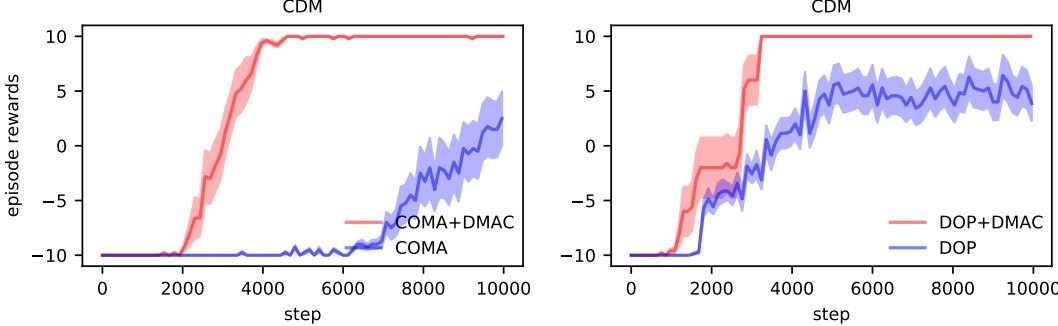

Figure 8: Learning curves in terms of mean episode rewards of COMA and DOP groups in the CDM environment used by the DOP paper.

code is based on the implementation of PyMARL (Samvelyan et al., 2019), MAAC (Iqbal and Sha, 2019), DOP (Wang et al., 2021b), FOP (Zhang et al., 2021) and an open source code for algorithms in SMAC (https://github.com/starry-sky6688/StarCraft).

## C.3 EXPERIMENT SETTINGS

We trained each algorithms for five runs with different random seeds. In SMAC tasks, we train each algorithm for one million time steps in each run for COMA, QMIX, MAAC, and DOP groups and two million timesteps for FOP groups. We do all the experiments by a server with 2 NVIDIA A100 GPUs.

## D ADDITIONAL RESULTS

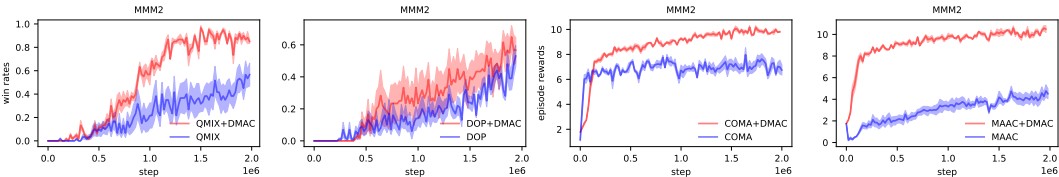

Figure 9: Learning curves of QMIX, DOP, COMA and MAAC groups on the map MMM2 in SMAC.

Figure 5 shows the learning curves in terms of cover rates of COMA, QMIX, MAAC and DOP groups in the randomly generated stochastic game.

Figure 6 shows the learning curves of COMA, MAAC, QMIX and DOP groups in terms of mean episode rewards in each SMAC map.

Figure 7 shows the learning curves of FOP+DMAC and FOP in terms of mean episode rewards in 3s_vs_3z, 2c_vs_64zg and MMM2.

Figure 8 shows the learning curves in terms of mean episode rewards of COMA and DOP groups in the CDM environment used by the DOP paper.

Figure 9 shows the learning curves of QMIX and DOP groups in terms of win rate, and COMA and MAAC groups in terms of episode reward on the map MMM2 in SMAC.

