# OpenReview forum: "Divergence-Regularized Multi-Agent Actor-Critic"
_ICLR.cc/2022/Conference — ICLR 2022 Submitted_

### Official Review · Reviewer_GUrq · 2021-11-01

**Correctness:** 3
**Technical Novelty And Significance:** 2
**Empirical Novelty And Significance:** 2
**Recommendation:** 5
**Confidence:** 4

**Main Review:**

### Strong points
1. In general, I think the proposed framework is at least new in the MARL literature. It is interesting to read a paper with a full deviation of such a divergence formulation of the learning objective.
2. The mathematical justifications are completely and sufficient for the purpose of presentation, although I do feel some of them could be possible shrank or moved to the appendix to save some space for more discussions and experiments. For example, Lemma 2 & 3 look just the same to me, and the discussions on applying DMAC to COMA look nothing special to the audience familiar with COMA.
3. I really appreciate seeing that the proposed methods can be really combined with multiple algorithms and gain improvement naturally.

### Weak points
1. Yes, the method is new in MARL but I'm still a bit concerned about the novelty of all the technical content.

There are sufficient mathematical discussions but I can still see most of the techniques are somewhat straightforward from existing works. For example, the value decomposition policy improvement theorem seems direct to obtain from the DOP paper; the mirror descent analysis seems very similar to the analysis in the DAPO paper. The authors state in the introduction section that "DAPO cannot be trivially extended to cooperative MARL settings". However, I don't really get this argument. It seems that all the mirror decent analysis simply holds directly with the assumption of linear-value decomposition. So, at least from the current form of the paper, the authors would have been done a better job explaining why those methods are challenging to be applied in MARL and what technical ideas really makes this paper distinct (rather simply saying that paper A is working on single-agent RL while paper B is on MARL). I hope the authors make more clarifications and discussions in the next revision.

2. Experiments are not sufficiently convincing.

(1) For Fig.1, why not run the same experiment setting as DOP so that we can have a straightforward comparison? From the curves in Fig.1, the behavior of COMA and DOP are contradicting the curves in the DOP paper.

(2) The paper utilizes COMA as its on-policy baseline. It is a fair choice, but it is been known that COMA has less satisfying performances in many domains. Is it possible to include some strong on-policy baselines like PPO or IMPALA?

(3) Except for FOP, the SMAC maps used in the experiments are less challenging. It would be more convincing if more hard maps can be examined. Also, for FOP, why not perform experiments on those simpler maps? The authors state that "COMA+DMAC has higher win rates than COMA in most cases at convergence" and "DOP+DMAC learns faster than DOP in most cases and finally converges to better performance". I agree that DMAC enhancement makes learning faster but I cannot really see the performances at convergence ---- at least from the visualization, DOP and COMA are still improving on some maps.


### Writing
1. It would be better if the authors could include some justifications at the beginning of section 4 on why such a ratio penalty is preferred. What is the motivation for this choice? Is this simply due to that we can eventually derive a nice KL-penalized value formulation?

2. "Besides, DMAC is **benefited** for exploration and stable policy improvement." --> beneficial

3. "**we** find that COMA+DMAC has higher win rates than COMA in most cases at convergence," --> capital w

**Summary Of The Paper:**

This paper suggests a DMAC learning framework by introducing a divergence penalty between the learning policy between a target policy into the RL framework instead of the commonly used entropy penalty. It can be shown that the proposed objective can be easily combined with various MARL algorithms and can eventually lead to the optimal policy of the underlying unmodified MDP with properly set target policies. The paper rigorously studies the mathematical foundation of the algorithm following a standard workflow and presents a few experiments showing that DMAC could improve the learning efficiency of various MARL algorithms.

**Summary Of The Review:**

This paper presents a new algorithmic framework with sufficient mathematical discussions, although I'm a bit concerned about its current form. However, the paper can be improved if more novelty discussions can be presented and more thorough experiments can be conducted.

---

> ### Author Response · Authors · 2021-11-16
> **Response to reviewer GUrq part 2**
>
> For your question 2.(2), our paper focuses on the cooperative MARL algorithms, so we think the single agent on-policy algorithms such as PPO and  IMPALA are out of scope.  Though COMA has less satisfying performances, it is the closest method to the policy gradient theorem in cooperative MARL settings. Moreover, we want to show the improvement to the existing methods when combining with DMAC and the initial performances of the existing methods are not such important. If you care more about the initial performance, we think our results of the DOP group are still convincing.
>
>
>
> For your question 2.(3), we run experiments on the super hard map MMM2 in SMAC for the 4 groups of methods. The results are illustrated in the Table 2 and Table 3. MMM2 is too hard for COMA and MAAC groups to win, so we compare their performance in terms of episode rewards. From the results in Table 2 and Table 3, we could find that DMAC also performs better than the baselines. *The learning curves are given in Appendix (Figure 9) in the revision.*
>
> **Table 2: The mean and standard deviation of win rates of QMIX and DOP groups  on the map MMM2 of SMAC . The results are from 5 random seeds.**
>
> | time steps | 0                | 0.4M             | 0.8M             | 1.2M             | 1.6M             | 2M               |
> | :--------- | ---------------- | :--------------- | :--------------- | :--------------- | :--------------- | :--------------- |
> | QMIX+DMAC  | 0.000 $\pm$0.000 | 0.050 $\pm$0.041 | 0.317 $\pm$0.085 | 0.767$\pm$0.103  | 0.967 $\pm$0.024 | 0.983$\pm$0.024  |
> | QMIX       | 0.000 $\pm$0.000 | 0.033$\pm$0.047  | 0.200 $\pm$0.248 | 0.317 $\pm$0.317 | 0.450 $\pm$0.308 | 0.567 $\pm$0.249 |
> | DOP+DMAC   | 0.000 $\pm$0.000 | 0.000 $\pm$0.000 | 0.233 $\pm$0.330 | 0.267 $\pm$0.342 | 0.383$\pm$0.259  | 0.617$\pm$0.184  |
> | DOP        | 0.000 $\pm$0.000 | 0.010$\pm$0.020  | 0.130 $\pm$0.140 | 0.180$\pm$0.160  | 0.290 $\pm$0.058 | 0.530 $\pm$0.166 |
>
> **Table 3: The mean and standard deviation of episode rewards of COMA and MAAC groups  on the map MMM2 of SMAC . The results are from 5 random seeds.**
>
> | time steps | 0                | 0.4M             | 0.8M             | 1.2M              | 1.6M             | 2M               |
> | :--------- | ---------------- | :--------------- | :--------------- | :---------------- | :--------------- | :--------------- |
> | COMA+DMAC  | 1.862 $\pm$0.131 | 8.645 $\pm$0.718 | 9.395 $\pm$0.656 | 9.740 $\pm$0.530  | 9.855 $\pm$0.426 | 9.753 $\pm$1.100 |
> | COMA       | 1.167 $\pm$0.238 | 6.999 $\pm$0.520 | 7.222 $\pm$0.560 | 7.197$\pm$0.833   | 6.804 $\pm$0.920 | 6.743 $\pm$1.005 |
> | MAAC+DMAC  | 1.958 $\pm$0.030 | 8.597 $\pm$0.271 | 9.486$\pm$0.348  | 10.075 $\pm$0.130 | 10.063$\pm$0.482 | 10.570$\pm$0.657 |
> | MAAC       | 1.715 $\pm$0.069 | 2.063$\pm$1.125  | 2.952$\pm$0.957  | 3.346$\pm$0.740   | 4.189$\pm$0.877  | 4.771$\pm$1.446  |
>
>
>
> As for the experiments of FOP, we have taken 3 SMAC tasks with 3 different levels from the FOP paper and DMAC performs better than FOP in all of these 3 SMAC tasks. We think these results are enough for our purpose of comparing DMAC and entropy regularization methods.
>
> We are sorry for the usage of 'convergence' in our statement. What we want to state is that COMA+DMAC and DOP+DMAC perform better at the end of the training. We have corrected this in the revision.
>
> As for the motivation of the divergence regularization, the first is that with our update rules, we could eliminate the bias caused by regularization for the converged policy as well as obtain an off-policy method with monotonic improvement guarantees.  *We have concluded these discussions about the target policy as Theorem 2 in Section 4.5 in the revision.* The second is that the divergence regularization encourages agents to take actions whose probability has decreased and discourages agents to take actions whose probability has increased, so it is beneficial to exploration which can also be witnessed from the empirical result in Figure 2.
>
>
>
> Please let us know if our responses have addressed your concerns or if you have additional comments/questions.

---

> ### Author Response · Authors · 2021-11-16
> **Response to reviewer GUrq part 1**
>
> We think DMAC is actually a general framework and it is unavoidable to absorb some existing ideas when combined with existing algorithms.  But we want to argue that the main contribution of DMAC is that DMAC is a naturally off-policy method, can be monotonically improved and is not biased by the regularizer at convergence. As for the difference between DMAC and DAPO, though the formulas of the policy gradients of both methods look similar, they are actually *different*. DAPO obtains its policy update rule from the perspective of the policy gradient theorem, which means the distribution of states is limited by the current policy $\pi$ and DAPO needs to update its policy with on-policy learning. We obtains the policy update rule of DMAC from a different perspective and the objective of the DMAC policy just needs to be optimized in each state. So the distribution of state is irrelevant  to  the current policy and we can update the policy of DMAC with off-policy learning. The influence of sample efficiency will be enlarged in the MARL settings. Moreover the off-policy correction trick V-trace used by DAPO is actually biased for the policy update. Finally, our empirical result of DAPO on the map 3m in SMAC can show that directly extending DAPO to MARL settings will not obtain good performance.
>
>
>
> For your question 2.(1), we need a stochastic game with more states to compare the exploration abilities of COMA and COMA+DMAC rather than the stateless CDM environment used by the DOP paper. Moreover, we think our stochastic game is more complicated than the CDM and more convincing.  As for the performance of COMA and DOP, we think the environment has been changed and the behavior of  two methods are not necessarily the same. We get the learning curves of DOP by running the open source code of the DOP paper. We have checked the results of DOP and find that the performance of DOP is not stable in our stochastic game and its performance is quite low in the 2 out of 5 seeds and is higher than COMA in the other 3 seeds. Finally, for a straightforward comparison, we run experiments in the CDM environment for the COMA and DOP groups and the results are illustrated in Table 1.  We could find both COMA+DMAC and DOP+DMAC perform better than the baselines and can achieve the optimal policy in the CDM (getting the highest return of 10.0).  *The learning curves are given in Appendix (Figure 8) in the revision.*
>
> **Table 1: The mean and standard deviation of returns of COMA and DOP groups in the CDM environment used by the DOP paper . The results are from 5 random seeds.**
>
> | time steps | 0                  | 2K                 | 4K                | 6K               | 8K                | 10K                   |
> | :--------- | ------------------ | :----------------- | :---------------- | :--------------- | :---------------- | :-------------------- |
> | COMA+DMAC  | -10.000 $\pm$0.000 | -10.000 $\pm$0.000 | -5.800 $\pm$5.879 | 1.000 $\pm$9.077 | 10.000 $\pm$0.000 | **10.000** $\pm$0.000 |
> | COMA       | -10.000 $\pm$0.000 | -10.000$\pm$0.000  | -10.000$\pm$0.000 | -9.750$\pm$0.559 | -2.500$\pm$9.803  | 2.000$\pm$10.954      |
> | DOP+DMAC   | -10.000 $\pm$0.000 | -2.000$\pm$9.798   | 10.000$\pm$0.000  | 10.000$\pm$0.000 | 10.000$\pm$0.000  | **10.000**$\pm$0.000  |
> | DOP        | -10.000$\pm$0.000  | -6.000$\pm$2.563   | 1.286$\pm$4.772   | 4.714$\pm$6.064  | 5.571$\pm$6.366   | 5.143$\pm$6.243       |

---

> ### Author Response · Authors · 2021-11-27
> **Follow-up**
>
> As the discussion will end soon, we would like to know whether our responses have addressed your concerns. Please let us know if you have further questions.

---

### Official Review · Reviewer_sxWU · 2021-11-01

**Correctness:** 3
**Technical Novelty And Significance:** 2
**Empirical Novelty And Significance:** 2
**Recommendation:** 5
**Confidence:** 3

**Main Review:**

Strengths:
Theoretical derivations have been included in the analysis of the algorithm in addition to the experiments.
Experiments conducted are comprehensive, covering most of the state-of-the-art algorithmic benchmarks.

Weaknesses:
The motivation for the target policy rho is not well-explained. What exactly is rho, and why do we need to maintain a rho to make learning more effective? Does it represent prior knowledge of team level policy pre-defined by the user or something else? If we already know what the optimal policy is, why do we need DMAC? If we don’t necessarily know the optimal policy, how can rho be chosen such that rho will help learning? Neither theoretical or empirical evaluations seem to address these concerns.
The degree of novelty seems unclear. On a high level, DMAC looks like a slight modification of the soft actor-critic(SAC) algorithm with joint action spaces, particularly in the definition of the objective function. The update rule is also standard in RL literature, and it would be ideal if the authors can further emphasize how their method differs from existing benchmarks.



**Summary Of The Paper:**

This paper proposes DMAC, a general off-policy actor-critic framework with entropy-based divergence regularization for multi-agent reinforcement learning to better coordinate multiple agents in a complicated environment. Theoretical derivations and empirical investigations on several state-of-the-art algorithmic benchmarks over the SMAC environment demonstrate the effectiveness of DMAC.

**Summary Of The Review:**

DMAC does provide improvements over existing algorithmic benchmarks, but the level of novelty seems limited as the DMAC method resembles SAC and several components of DMAC has questionable motivation. I’d be happy to change my score, if the authors address these issues accordingly.

---

> ### Author Response · Authors · 2021-11-16
> **Response To Reviewer sxWU**
>
> We have discussed about the target policy $\rho$ in the Section 4.5.  In our paper, the definition of $\rho$ is from the iteration
> $$
> \boldsymbol{\pi}^{t+1} = \arg \max_{{\boldsymbol{\pi}}} \sum_{s,\boldsymbol{a}} \mu_{\boldsymbol{\pi}}(s,\boldsymbol{a}) r(s,\boldsymbol{a})  - \lambda D_{\operatorname{C}} \left( \mu_{\boldsymbol{\pi}} \| \mu_{{\boldsymbol{\pi}}^t} \right).
> $$
> We take $\rho$ as the past policy $\pi^t$ in the process of optimizing the policy $\pi^{t+1}$.  The theory of mirror descent will guarantee this iteration to be converged. When the policy has converged, the extra part  $D_{\operatorname{C}} \left( \mu_{\boldsymbol{\pi}^{t+1}} \| \mu_{{\boldsymbol{\pi}}^t} \right)$ will converge to zero. With this definition of  $\rho$, we actually eliminate the bias caused by the regularization for the converged policy.  Moreover , with the definition of $D_{\operatorname{C}} \left( \mu_{\boldsymbol{\pi}} \| \mu_{{\boldsymbol{\rho}}} \right)$, we have $D_{\operatorname{C}} \left( \mu_{\boldsymbol{\pi}} \| \mu_{{\boldsymbol{\rho}}} \right) \ge 0$. Then we can obtain the following inequalities:
> $$
> J(\boldsymbol{\pi}^{t + 1}) \ge J(\boldsymbol{\pi}^{t + 1}) - \lambda D_{\operatorname{C}} \left( \mu_{\boldsymbol{\pi}^{t+1}} \| \mu_{{\boldsymbol{\pi}}^t} \right) \ge J(\boldsymbol{\pi}^{t }) - \lambda D_{\operatorname{C}} \left( \mu_{\boldsymbol{\pi}^{t}} \| \mu_{{\boldsymbol{\pi}}^t} \right) = J(\boldsymbol{\pi}^{t }),
> $$
> The second inequality is from the definition of $\boldsymbol{\pi}^{t + 1}$. This conclusion means the policy sequence obtained by this iteration improves monotonically in the original MDP. *We have concluded these discussions about the target policy as Theorem 2 in Section 4.5 in the revision.* Also, the regularizer will be beneficial to explorations from our analysis in the Section 4.5 which could be also witnessed from the empirical results in Figure 2.  As for the optimal policy, DMAC does not need the optimal policy or other prior knowledge. Actually, DMAC provides a method with monotonic improvement which tries to find the optimal policy.
>
> Note that this iteration indicates we need to fix $\rho$ until we find the optimal policy $\pi^{t+1}$ to complete one iteration from $\pi^t$ to $\pi^{t+1}$, which however is not efficient. So in practice we take $\rho$ as the moving average of $\pi$ as an approximation.  Such an approximation is effective, verified by our empirical results.
>
> As for the difference between DMAC and SAC, we introduce the target policy $\rho$ and replace the entropy regularization with divergence regularization.  With our update rule of target policy,  we can eliminate the bias of the converged policy caused by the entropy regularization.  We analyze the divergence-regularized MDP and obtain naturally off-policy update rules for the critic and policies with monotonic improvement guarantee. As mentioned in the paper some of our proofs refer to the results of SAC, so some conclusions look similar. The empirical results in Figure 4 (compared with FOP) can be an evidence for the advantages of DMAC and the bias of entropy regularization.
>
> Please let us know if the responses above have addressed your comments or if you have additional comments/questions.

---

> ### Author Response · Authors · 2021-11-27
> **Follow-up**
>
> As the discussion will end soon, we would like to know whether our responses have addressed your concerns. Please let us know if you have further questions.

---

### Official Review · Reviewer_b5Li · 2021-11-02

**Correctness:** 3
**Technical Novelty And Significance:** 2
**Empirical Novelty And Significance:** 4
**Recommendation:** 6
**Confidence:** 3

**Main Review:**

**Strengths:**
1. The paper is well written, organized, and explains the main insights well.
2. The proposed framework has a very appealing and important benefit that DMAC is flexible and can be easily combined with previous works. This benefit is highlighted well by combining with four representative MARL algorithms in the challenging SMAC domain.
3. DMAC makes a conclusion in a principled manner, where the benefit of DMAC is thanks to the improved exploration without introducing the bias (Section 4.5).

**Questions:**
1. As pointed in the paper, the theoretical study in Section 4.2 is assuming the fixed target policy. However, in practice, the target joint policy is non-stationary/updated (see Algorithm 1). Hence, I wonder whether the theoretical conclusions (i.e., monotonic improvement, optimal policy convergence) would still hold with the non-stationary target joint policy.
2. To address the partial observability, an RNN layer is used. However, it is unclear from reading the main paper and appendix, how the hidden state is handled in the off-policy implementation. Is the hidden state stored in the replay buffer as in Hausknecht and Stone, 2015?
3. What are the possible future directions of DMAC? Can DMAC be extended to general-sum games (other than cooperative settings), and if not, what are the expected challenges?

**Reference:**
Matthew Hausknecht, Peter Stone. Deep Recurrent Q-Learning for Partially Observable MDPs. AAAI, 2015.

**Summary Of The Paper:**

This paper develops a new multiagent framework with regulated divergence to address the learning of biased policies in the maximum-entropy approach. Specifically, the main contributions include the developments of the divergence policy iteration for general cooperative MARL settings and the off-policy divergence regularized multiagent actor-critic framework (DMAC). Empirical results show that DMAC can improve the performance of existing MARL frameworks in various multiagent domains.

**Summary Of The Review:**

Initially, I vote for a score of 6. While some parts of the theoretical studies are based on related works (e.g., SAC, DOP), DMAC shows the appealing benefit that the framework can be easily combined with prior MARL works, so it has much applicability. After reading the authors' responses to my questions, I am open to raising my score.

---

> ### Author Response · Authors · 2021-11-16
> **Response To Reviewer b5Li**
>
> The practical method of DMAC is actually from the following iteration
> $$
> \boldsymbol{\pi}^{t+1} = \arg \max_{{\boldsymbol{\pi}}} \sum_{s,\boldsymbol{a}} \mu_{\boldsymbol{\pi}}(s,\boldsymbol{a}) r(s,\boldsymbol{a})  - \lambda D_{\operatorname{C}} \left( \mu_{\boldsymbol{\pi}} \| \mu_{{\boldsymbol{\pi}}^t} \right).
> $$
> This iteration means we need to fix $\rho = \pi^t$ and use the Divergence Policy Iteration to find the optimal policy as $\pi^{t + 1}$ which is too slow especially for MARL settings.  So we use two approximations in  Algorithm 1:  one is replacing the Divergence Policy Iteration with one gradient step, the other is taking $\rho$ as the moving average of $\pi$.  The theoretical conclusions will not hold in the practical method with the approximation of $\rho$.  But we have tried to make the approximation of $\rho$ be close to the ideal condition needed by the theoretical conclusions.  The moving average of $\pi$ will also eliminate the bias caused by regularization at convergence and the difference between $\rho^t$ and $\rho^{t + 1}$ will be very small with an appropriate hyperparameter $\tau$ which will be close to the stable $\rho$ needed by the gradient update of the policy. *Moreover, we have concluded the discussions about the target policy as Theorem 2 in Section 4.5 in the revision.*
>
>
>
> As for the hidden states of the RNN, in practice, we just sample several episodes from the replay buffer and calculate the hidden states for each episode from the first step to the last step. We do not save the hidden states in the replay buffer.  Our process of the hidden states just follows the implementations of previous cooperative MARL algorithms as in PyMARL (Samvelyan et al., 2019).
>
>
>
> Extending DMAC to general-sum games is a good future direction.  The main challenge may be how to handle different objectives of agents. The current theoretical results of DMAC is only for cooperative settings.  Some further study of DMAC with different objectives need to be done in the general-sum game settings.

---

> > ### Comment · Reviewer_b5Li · 2021-11-27
> > **Rebuttal Response**
> >
> > I would like to thank the authors for their detailed response and for making the changes to the paper accordingly. I have read the rebuttal and it clarified my questions. I agree with the other reviewers' concern on the novelty side, but I also value this paper's empirical results. Hence, I am leaning towards keeping my score. If the authors will open-source the method, this can further strengthen the empirical contribution.

---

> > > ### Author Response · Authors · 2021-11-29
> > > **Follow-up**
> > >
> > > Thanks for your positive feedback. We will certainly release the code if the paper is accepted.

---

### Official Review · Reviewer_nt66 · 2021-11-02

**Correctness:** 3
**Technical Novelty And Significance:** 3
**Empirical Novelty And Significance:** 2
**Recommendation:** 6
**Confidence:** 2

**Main Review:**

Regarding related work, it seems that closely related is Wang et al. 2019. I propose the give more details on the differences to the current work, as to why it cannot be applied to the cooperative MARL setting the authors study (because intuitively, cooperative MARL is verly close to the single-agent case).

Am I understanding correctly that the theoretical guarantees (monotonic improvement) only hold for a fixed target policy rho? I.e., not for the additional update rule for rho in 4.5?  If so, that would be OK, it is clear that one cannot understand everyhting in one paper. However, it may be worth making more explicit if there is a theoretical guarantee also for this general method or not, and if there in fact is, then explicitly state this as a theorem.

I suggest to give a bit more overview and motivation of what is being done where in the paper and how things are connected. For instance, it remained unclear to me how rho is defined, or what is intuitively is, until sec 4.5.

**Summary Of The Paper:**

For cooperative multi-agent reinforcement learning, they propose an off-policy(?) learning method that is reqularized by the KL divergence to some target policy, which, in contrast to classic max entropy RL, allows to circumvent(?) the bias that the regularizer introduces (eq18).
They give several special cases / tricks of how the learning can be somewhat decoupled between the agent (Lem 3, end of sec4.4). They paper comprises theory about convergence of the method, as well as experiments in toy envs.

**Summary Of The Review:**

MARL is a very challenging task, and I agree that entropy regularization introduces problematic biases, so they address a relevant and challenging problem. The paper is strong in presenting both theory (I took a brief look at the proofs, the authors seem to know what they are doing, but I did not go through details) and experiments.

What I'm not so sure about is how deep the contribution is compared to previous work, in particular Wang et al. 2019, or, for instance, (Haarnoja et al., 2018), which is cited in the proof of Lem 2. Additionally, it seems the guarantees are only for fixed rho, while the full strength (nonbiasedness) of the method seems to come into play only when updating rho as well.

---

> ### Author Response · Authors · 2021-11-16
> **Response To Reviewer nt66**
>
> The difference between DMAC and DAPO is discussed in the Section 4.4. DAPO obtains its update rule of policy from the perspective of the policy gradient theorem, which means the distribution of states is limited by the current policy $\pi$ and DAPO needs to update its policy with **on-policy learning**. We obtains the policy update rule of DMAC from a different perspective and the objective of the DMAC policy just needs to be optimized in each state. So the distribution of state is irrelevant  to  the current policy and we can update the policy of DMAC with **off-policy learning**.
>
>
>
> In the Section 4.5, we actually propose an ideal version of DMAC:
> $$
> \boldsymbol{\pi}^{t+1} = \arg \max_{{\boldsymbol{\pi}}} \sum_{s,\boldsymbol{a}} \mu_{\boldsymbol{\pi}}(s,\boldsymbol{a}) r(s,\boldsymbol{a})  - \lambda D_{\operatorname{C}} \left( \mu_{\boldsymbol{\pi}} \| \mu_{{\boldsymbol{\pi}}^t} \right).
> $$
> In this ideal version, we take the target policy as $\rho = \pi^t$, use the Divergence Policy Iteration to find the optimal policy as $\pi^{t + 1}$ given this fixed $\rho$ and repeat this iteration.  The convergence of this iteration is guaranteed by the theory of the mirror descent and the converged policy will not be biased by the regularization.  *We have concluded these discussions about the target policy as Theorem 2 in Section 4.5 in the revision.* All of our theoretical guarantees hold for this ideal version. But this ideal version is too slow and we use two approximations in the practical method in Algorithm 1.  The practical method of DMAC will not satisfy the condition of the theoretical guarantees.
>
> The reason of taking the moving average of $\pi$ as the approximation of $\rho$ is two-fold.  One reason is that the moving average can also represent the past policy and will also eliminate the bias caused by the regularization when the policy $\pi$ has converged. The other reason is that the difference between $\rho^t$ and $\rho^{t + 1}$ can be controlled by the hyperparameter $\tau$ to be close to the stable $\rho$ condition needed by the gradient update of the policy.
>
> The main difference between DMAC and SAC also lies in the target policy $\rho$. We introduce the target policy $\rho$ and  utilize our update rule of target policy to eliminate the bias caused by the regularization. So DMAC is able to maintain the property of off-policy learning and monotonic improvement and avoid the bias caused by regularization at the same time.

---

### Author Response · Authors · 2021-11-16
**Summary**

**Summary**: Thank the reviewers for their positive assessment and helpful suggestions for improvement. The main contribution of this paper is DMAC, *a flexible framework that can be combined with many existing algorithms and improve their performance*. Some comments from the reviewers are particularly important to make the paper clearer, including  the motivation of the target policy $\boldsymbol{\rho}$, the approximation used in our practical method and the difference between DMAC and previous work such as DAPO and SAC. The responses to these comments are provided as follows, for the reviewers and readers, to fully understand the merits of this paper.

- **The motivation of the target policy $\boldsymbol{\rho}$.** In our paper, the definition of $\rho$ is from the iteration
  $$
  \boldsymbol{\pi}^{t+1} = \arg \max_{{\boldsymbol{\pi}}} \sum_{s,\boldsymbol{a}} \mu_{\boldsymbol{\pi}}(s,\boldsymbol{a}) r(s,\boldsymbol{a})  - \lambda D_{\operatorname{C}} \left( \mu_{\boldsymbol{\pi}} \| \mu_{{\boldsymbol{\pi}}^t} \right).
  $$
  We take $\rho$ as the past policy $\pi^t$ in the process of optimizing the policy $\pi^{t+1}$.  The theory of mirror descent will guarantee this iteration to be converged. When the policy has converged, the extra part  $D_{\operatorname{C}} \left( \mu_{\boldsymbol{\pi}^{t+1}} \| \mu_{{\boldsymbol{\pi}}^t} \right)$ will converge to zero. With this definition of  $\rho$, we actually eliminate the bias caused by the regularization for the converged policy.  Moreover , with the definition of $D_{\operatorname{C}} \left( \mu_{\boldsymbol{\pi}} \| \mu_{{\boldsymbol{\rho}}} \right)$, we have $D_{\operatorname{C}} \left( \mu_{\boldsymbol{\pi}} \| \mu_{{\boldsymbol{\rho}}} \right) \ge 0$. Then we can obtain the following inequalities:
  $$
  J(\boldsymbol{\pi}^{t + 1}) \ge J(\boldsymbol{\pi}^{t + 1}) - \lambda D_{\operatorname{C}} \left( \mu_{\boldsymbol{\pi}^{t+1}} \| \mu_{{\boldsymbol{\pi}}^t} \right) \ge J(\boldsymbol{\pi}^{t }) - \lambda D_{\operatorname{C}} \left( \mu_{\boldsymbol{\pi}^{t}} \| \mu_{{\boldsymbol{\pi}}^t} \right) = J(\boldsymbol{\pi}^{t }),
  $$
  The second inequality is from the definition of $\boldsymbol{\pi}^{t + 1}$. This conclusion means the policy sequence obtained by this iteration improves monotonically in the original MDP. *We have concluded these discussions about the target policy as Theorem 2 in Section 4.5 in the revision.* Also, the regularizer encourages agents to take actions whose probability has decreased and discourages agents to take actions whose probability has increased, so it is beneficial to exploration which can also be witnessed from the empirical result in Figure 2.

- **The approximation used in our practical method.** We have used two approximations in  our practical method:  one is replacing the Divergence Policy Iteration with one gradient step, the other is taking $\rho$ as the moving average of $\pi$. We have tried to make the approximation of $\rho$ be close to the ideal condition needed by the theoretical conclusions.  The moving average of $\pi$ will also eliminate the bias caused by regularization at convergence and the difference between $\rho^t$ and $\rho^{t + 1}$ will be very small with an appropriate hyperparameter $\tau$ which will be close to the stable $\rho$ needed by the gradient update of the policy.

- **The difference between DMAC and previous works.** The difference between DMAC and DAPO is discussed in the Section 4.4. DAPO obtains its update rule of policy from the perspective of the policy gradient theorem, which means the distribution of states is limited by the current policy $\pi$ and DAPO needs to update its policy with **on-policy learning**. We obtains the policy update rule of DMAC from a different perspective and the objective of the DMAC policy just needs to be optimized in each state. So the distribution of state is irrelevant  to  the current policy and we can update the policy of DMAC with **off-policy learning**.

  The main difference between DMAC and SAC lies in the target policy $\rho$. We introduce the target policy $\rho$ and  utilize our update rule of target policy to eliminate the bias caused by the regularization. So DMAC is able to maintain the property of off-policy learning and monotonic policy improvement, and avoid the bias caused by regularization at the same time.

---

### Decision · Program_Chairs · 2022-01-20

**Decision:**

Reject

**Comment:**

The paper presents multi-agent RL framework that uses the divergence between the learned policies and a target policy as a penalty that pushes the agent to learn cooperative strategies. The proposed method is built on top of an existing one (DAPO, Wang et al., 2019). Empirical experiments clearly show the advantage of the proposed method.

The reviews for this paper are mixed and borderline. The reviewers appreciate the experiments reported in the paper and that indicate the advantage of the proposed method. But two reviewers do not think that the proposed analysis is sufficiently novel compared to an existing one (DAPO). The responses provided by the authors were appreciated, but did not dissipate these concerns.